# A Snapshot of Influence: A Local Data Attribution Framework for Online Reinforcement Learning

**Yuzheng Hu**[*]
UIUC
Urbana, IL 61801
yh46@illinois.edu

**Fan Wu**[*]
UIUC
Urbana, IL 61801
fanw6@illinois.edu

**Haotian Ye**
Stanford University
Stanford, CA 94305
haotianye@stanford.edu

**David Forsyth**
UIUC
Urbana, IL 61801
daf@illinois.edu

**James Zou**
Stanford University
Stanford, CA 94305
jamesz@stanford.edu

**Nan Jiang**
UIUC
Urbana, IL 61801
nanjiang@illinois.edu

**Jiaqi W. Ma**[†]
UIUC
Urbana, IL 61801
jiaqima@illinois.edu

**Han Zhao**[†]
UIUC
Urbana, IL 61801
hanzhao@illinois.edu

## Abstract

Online reinforcement learning (RL) excels in complex, safety-critical domains but suffers from sample inefficiency, training instability, and limited interpretability. Data attribution provides a principled way to trace model behavior back to training samples, yet existing methods assume fixed datasets, which is violated in online RL where each experience both updates the policy and shapes future data collection. In this paper, we initiate the study of data attribution for online RL, focusing on the widely used Proximal Policy Optimization (PPO) algorithm. We start by establishing a *local* attribution framework, interpreting model checkpoints with respect to the records in the recent training buffer. We design two target functions, capturing agent action and cumulative return respectively, and measure each record's contribution through gradient similarity between its training loss and these targets. We demonstrate the power of this framework through three concrete applications: diagnosis of learning, temporal analysis of behavior formation, and targeted intervention during training. Leveraging this framework, we further propose an algorithm, iterative influence-based filtering (IIF), for online RL training that iteratively performs experience filtering to refine policy updates. Across standard RL benchmarks (classic control, navigation, locomotion) to RLHF for large language models, IIF reduces sample complexity, speeds up training, and achieves higher returns. Together, these results open a new direction for making online RL more interpretable, efficient, and effective.

## 1 Introduction

Reinforcement learning (RL) has achieved remarkable success across a wide range of decision-making tasks, from game playing [Mnih et al., 2015, Silver et al., 2016] to robotic control [Andrychowicz et al., 2020] and the alignment of large language models (LLMs) [Ouyang et al., 2022]. Among its

---

[*]Equal contribution [†]Equal advising

39th Conference on Neural Information Processing Systems (NeurIPS 2025).

variants, online RL, which continuously alternates between data collection and policy updates (e.g., A3C [Mnih et al., 2016], PPO [Schulman et al., 2017]), is well-suited to real-time, adaptive, and safety-critical domains such as autonomous driving, as it enables on-the-fly correction of mistakes and rapid adaptation to non-stationary environments [Sallab et al., 2017, Andrychowicz et al., 2020]. However, modern online RL faces several challenges, including sample inefficiency, high variance, and training instability, often requiring millions of interactions for convergence and yielding inconsistent performance across runs [Henderson et al., 2018, Yu, 2018, Dulac-Arnold et al., 2019].

These challenges, together with their deployment in high-stakes domains, necessitate a deeper understanding of the operational mechanisms of online RL. To this end, prior work has explored various methods for RL interpretability [Milani et al., 2024, Cheng et al., 2025]. While useful, these methods often lack the fine-grained explanations necessary for effective interventions or have limited applicability (see Sec. 6 for a detailed review of related work). Addressing these limitations requires exploring new paradigms.

In recent years, *data attribution* [Deng et al., 2025] has emerged as a powerful approach for machine learning interpretability, offering a complementary perspective by tracing model behaviors back to training data. This framework further benefits downstream applications such as data selection [Xia et al., 2024], bias mitigation [Wang et al., 2024], fact tracing [Chang et al., 2025], among others. However, applying data attribution to online RL is non-trivial. In online RL, agents continuously interact with their environment; each collected experience not only contributes to policy updates but also influences future rollouts collected by the evolving policy. This violates the core assumptions of traditional data attribution methods, which are designed for static datasets and fixed objectives.

In this work, we address this gap by presenting the first study of data attribution for online RL, specifically focusing on the widely used Proximal Policy Optimization (PPO) algorithm [Schulman et al., 2017]. Our contributions are threefold:

1. **A principled and flexible framework (Sec. 3).** We propose a local data attribution framework for online RL, interpreting model checkpoints w.r.t. the records from the recent training buffer. We define the attribution entity as the atomic unit in PPO training, design two target functions that capture agent actions and cumulative returns, and measure each record's influence through gradient similarity between its training loss and the target.

2. **Fresh insights into learning (Sec. 4).** We demonstrate the power of our framework through three applications: a) *diagnosis of learning*: we show records most harmful for learning feature inaccurate advantage estimates; b) *temporal analysis of behavior formation*: we reveal an intriguing phase transition of critical records in shaping agent behaviors; c) *targeted intervention*: we show that removing records with the most negative influences can effectively improve model training.

3. **Improved training (Sec. 5).** Building on the targeted intervention, we further develop an iterative influence-based filtering algorithm (IIF) that significantly improves standard online RL training. Across standard RL benchmarks to modern RLHF for large language models, IIF consistently improves *sample efficiency*, reduces *computational cost*, and enhances *final performance*.

## 2 Preliminaries

### 2.1 Online reinforcement learning

We consider the online RL setting, where an agent learns to maximize long-term returns by interacting with the environment. The environment $\mathcal{E}$ is modeled as a Markov Decision Process (MDP) defined by the tuple $(\mathcal{S}, \mathcal{A}, P, R, \gamma, d_0)$, where $\mathcal{S}$ is the state space, $\mathcal{A}$ the action space, $P$ the transition function, $R$ the reward function, $\gamma \in [0, 1]$ the discount factor, and $d_0 \in \mathcal{P}(\mathcal{S})$ the initial state distribution. At timestep $t$, the agent observes $s_t$, takes action $a_t$, receives reward $r_t$, and transitions to $s_{t+1}$.

Online RL typically proceeds in alternating **training rounds** of data collection and model training (Fig. 1). In round $k$, the data collection phase involves the agent executing the current policy $\pi_{\theta^{(k)}}$, sampling experiences over multiple episodes to accumulate $n$ transition records in a rollout buffer $B^{(k)}$. Each record contains the raw *transition* $(s_t, a_t, r_t)$ and several computed quantities, including the action log probability $\log \pi_{\theta^{(k)}}(a_t|s_t)$, estimated value $v_t$, and advantage estimate $\hat{A}_t$. Model parameters are then updated iteratively starting from $\theta_0^{(k)} = \theta^{(k)}$: at optimization step $j$, training on the mini-batch $\mathcal{B}_j^{(k)}$ drawn from $B^{(k)}$ updates parameters from $\theta_j^{(k)}$ to $\theta_{j+1}^{(k)}$. In this paper,

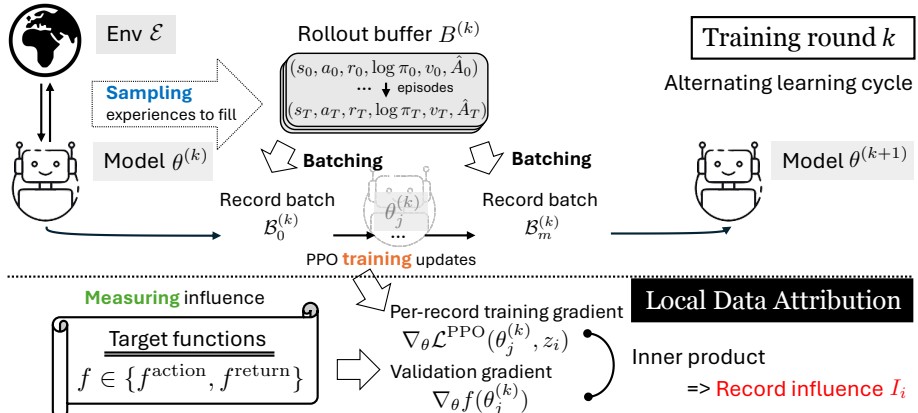

Figure 1: An Illustration of the alternating learning cycle in online RL (Sec. 2.1) and our local data attribution framework (Sec. 3.1). Online RL operates in alternating rounds of data collection and policy updates; our local data attribution framework quantifies how *individual* records from a single round influence different aspects of policy update in that round.

we focus on Proximal Policy Optimization (PPO), a widely used, effective algorithm in various applications [Berner et al., 2019, Andrychowicz et al., 2020, Ouyang et al., 2022].

**Proximal policy optimization (PPO) [Schulman et al., 2017].** PPO is a policy gradient method for online RL that optimizes a clipped surrogate function. The core PPO objective, which is typically combined with a value function loss and an entropy bonus during optimization, is defined as:

$$\mathcal{L}^{\mathrm{PPO}}(\theta) = \mathbb{E}_{(s,a) \sim \mathcal{B}_j^{(k)}} \left[ \min \left( \frac{\pi_\theta(a|s)}{\pi_{\theta^{(k)}}(a|s)} \hat{A}(s,a), \mathrm{clip} \left( \frac{\pi_\theta(a|s)}{\pi_{\theta^{(k)}}(a|s)}, 1 - \epsilon, 1 + \epsilon \right) \hat{A}(s,a) \right) \right],$$

where $\epsilon$ is a hyperparameter that limits policy changes between rounds and promotes stable learning.

## 2.2 Data attribution

Data attribution, which quantifies the influence of individual training samples on model behavior, has become increasingly important in machine learning [Grosse et al., 2023, Wang et al., 2023, Zheng et al., 2024]. Common techniques include influence functions [Koh and Liang, 2017], Data Shapley [Ghorbani and Zou, 2019], SGD-influence [Hara et al., 2019], TracIn [Pruthi et al., 2020], and TRAK [Park et al., 2023]. We focus on TracIn due to its conceptual simplicity, relative efficiency, and widespread use in recent works [Xie et al., 2024, Xia et al., 2024, Lin et al., 2024].

**TracIn [Pruthi et al., 2020].** TracIn measures the cumulative change in a *target function* $f(\theta)$ resulting from the optimization steps involving a specific training sample $z_i$. Formally, consider training a model parameterized by $\theta$ on a training set $\{z_i\}_{i=1}^n$ by minimizing the empirical loss $\sum_{i=1}^n \ell(\theta, z_i)$ using stochastic gradient descent (SGD). At step $j$, with parameters $\theta_j$, learning rate $\eta_j$, and mini-batch $\mathcal{B}_j$, a first-order Taylor expansion of $f(\theta)$ around $\theta_j$ gives:

$$f(\theta_j) - f(\theta_{j+1}) \approx \nabla_\theta f(\theta_j) \cdot (\theta_j - \theta_{j+1}) = \eta_j \sum_{i \in \mathcal{B}_j} \nabla_\theta f(\theta_j) \cdot \nabla_\theta \ell(\theta_j, z_i).$$

Accumulating these contributions over the relevant training iterations yields the TracIn score for $z_i$:

$$\mathrm{TracIn}(z_i) = \sum_{j: z_i \in \mathcal{B}_j} \eta_j \nabla_\theta f(\theta_j) \cdot \nabla_\theta \ell(\theta_j, z_i).$$

# 3 A Local Data Attribution Framework for Online RL

Online RL presents unique challenges for data attribution, due to the way data interacts with model parameters during learning. To tackle this challenge, we introduce a *local* attribution framework tailored to *local* policy optimization inherent in online RL.

**Challenges.** The key feature of online RL is *the circular dependency between data and model—* earlier experiences drive policy updates, and updated policies produce new experiences to learn

from. The dependency of data on model (red arrows in Fig. 2) is unique to online RL and cannot be addressed by existing attribution methods. Current data attribution methods include *retraining-based* (e.g., Ghorbani and Zou [2019]) and *gradient-based*, with the latter further divided into *static* and *dynamic* [Hammoudeh and Lowd, 2024]. Retraining-based methods require training the model once for each of the records being evaluated, which is computationally expensive in any setting and particularly prohibitive in RL. Static methods implicitly assume model parameters are obtained from solving an empirical risk minimization problem over a fixed dataset, which is violated in the non-stationary, sequential data setting here. While dynamic methods (e.g., TracIn)

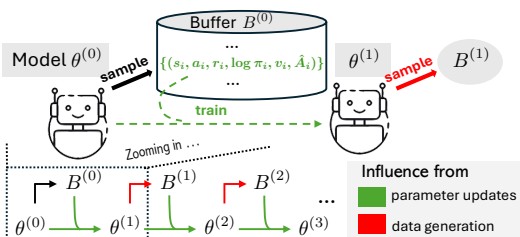

Figure 2: Twofold data influence: driving policy updates, shaping future data collection.

capture the temporal dependencies of training data influences on model parameters, they still fail to account for this key effect of *data-model dependency*. If we compute influence scores using the original formulas from standard supervised learning, they capture only the impact on parameter updates, ignoring the extra *channel* of influences through future data generation. As a result, the scores may deviate significantly from the true influence we seek to measure. Furthermore, quantifying influences through this channel is challenging because sampling is stochastic and non-differentiable.

## 3.1 A framework of local data attribution

Our local data attribution framework addresses the circular data-model dependency. Online RL involves a *local policy optimization* structure, i.e., round $k$ optimizes on a fixed buffer $B^{(k)}$ of on-policy data. Thus, each round serves as a natural unit of analysis. Our framework operates at this level, examining how records in $B^{(k)}$ contributes to the updates from $\theta^{(k)}$ to $\theta^{(k+1)}$. This circumvents the challenges in tracing influence through the complex, cascading, and non-differentiable dependencies across the training history. Below, we detail the three key components of our framework.

**Entity of attribution.** We consider attribution to individual training records in the rollout buffer, $z_i = (s_i, a_i, r_i, \log \pi_i, v_i, \hat{A}_i)$, collected from the environment using the current policy $\theta^{(k)}$. These records form the *atomic* unit used in PPO updates and provide a natural granularity for attribution.

**Target functions.** Training data influence is usually reflected through the impact on model behaviors. Here we focus on two core aspects of an RL agent: agent action and cumulative return.

*Agent action*: To identify records influencing the agent's decision to take a specific action $a$ at state $s$, we define a straightforward target function:

$$f^{\text{action}}(\theta) := \log \pi_\theta(a \mid s).$$

*Cumulative return*: We aim to understand which experience records contribute positively or negatively to the agent's ability to maximize cumulative return. Formally, the ideal quantity is the expected return $J(\theta) = \mathbb{E}_{\tau \sim \pi_\theta}[R(\tau)]$, where $R(\tau) = \sum_{t=0}^{T-1} r_t$ and trajectories $\tau$ are sampled by executing $\pi_\theta$. However, using $J(\theta)$ directly poses two fundamental challenges. *First*, unlike supervised learning with a fixed validation set, the data distribution in online RL is inherently policy-dependent. This intertwining of policy and evaluation means no fixed, universal validation set exists. *Second*, raw returns $R(\tau)$ exhibit high variance, leading to noisy influence estimates.

To address these challenges, we introduce a stable surrogate objective based on a reference policy $\pi^{\text{ref}}$ and advantage estimates $\hat{A}^{\text{ref}}$:

$$f^{\text{return}}(\theta) := \mathbb{E}_{\tau \sim \pi^{\text{ref}}, (s,a) \sim \tau} \left[ \log \pi_\theta(a \mid s) \hat{A}^{\text{ref}}(s, a) \right].$$

This target function is structurally equivalent to the objective of REINFORCE with a baseline [Sutton and Barto, 2018, Section 13.4]. By sampling from $\pi^{\text{ref}}$, we obtain a fixed evaluation distribution; using advantage estimates significantly reduces variance compared to raw returns. Maximizing $f^{\text{return}}(\theta)$ encourages increasing the probability of better-than-average actions and decreasing worse-than-average ones, capturing the essence of improving expected return while being tractable.

For attribution in round $k$, we set the reference policy $\pi^{\text{ref}} = \pi_{\theta^{(k)}}$, i.e., the policy snapshot at the beginning of the round. This is a key design choice of our *contextual* framework, which enables

us to ask: *For the agent at its current stage of training, which experiences will be most helpful or harmful for the next update?* Unlike a fixed, off-distribution reference that may provide misleading signals due to mismatch with the agent's current state, our dynamic reference evolves with training, providing a stable and relevant basis for meaningful evaluation and attribution. Furthermore, since the training rollout buffer $B^{(k)}$ is collected under $\pi_{\theta^{(k)}}$, we can directly use it as the validation dataset. We provide further discussions on this design choice in Sec. 4.3 and Sec. 5.1.

We note that one key contribution in our framework is the design of *tractable yet meaningful* target functions, particularly $f^{\text{return}}$, which can be reused in future work with alternative attribution methods.

**Remark 1** (Use cases of the two target functions). *The two target functions have different use cases. $f^{action}$ is mainly for diagnosis: understanding why the agent takes a specific action at a specific state (Sec. 4.2). On the other hand, $f^{return}$ assesses contribution to overall performance, which makes it suitable for both analysis (Sec. 4.1) and algorithmic policy improvement (Sec. 5).*

**Method of attribution.**  We adapt TracIn to our online RL setting. For record $z_i$ in the rollout buffer $B^{(k)}$, we compute its *influence score* by summing over the optimization steps $j$ within round $k$:

$$I_i := \sum_{j:z_i \in \mathcal{B}_j^{(k)}} \left\langle \nabla_\theta f(\theta_j^{(k)}), \nabla_\theta \mathcal{L}^{\text{PPO}}(\theta_j^{(k)}, z_i) \right\rangle, \quad \text{where } f \in \left\{ f^{\text{action}}, f^{\text{return}} \right\}.$$

Here, $\nabla_\theta f(\theta_j^{(k)})$ is the gradient of the target function evaluated at $\theta_j^{(k)}$, and $\nabla_\theta \mathcal{L}^{\text{PPO}}(\theta_j^{(k)}, z_i)$ is the per-sample gradient of the PPO training objective for record $z_i$. We also discuss two design choices in Sec. 5.1 which substantially reduce the computational and storage costs of the vanilla TracIn.

Finally, we clarify how to interpret the computed influence scores. Records with positive influence *benefit* behavior formation or learning, whereas those with negative influence *harm* it. We refer to records with the most positive influence as *top records* and those with the most negative influence as *bottom records*; these terms will be used throughout the remainder of the paper.

**Remark 2** (Extension to other online RL algorithms). *While we focus on PPO in our study, our framework readily extends to other online RL algorithms. For on-policy methods[2] such as TRPO [Schulman et al., 2015] and A3C [Mnih et al., 2016], the adaptation only requires modifying the per-sample loss gradient. For offline methods like DQN [Mnih et al., 2013], we need to additionally change the target function to the Bellman error. In all cases, our attribution framework reveals whether training records help or hinder learning at the agent's current state. A key distinction is that, on-policy methods allow direct validation with current data, whereas off-policy methods require sampling fresh rollouts.*

# 4   Applications of Local Data Attribution

We now illustrate the practical value of our framework. The framework delivers fresh insights for RL researchers and practitioners, enabling key applications such as diagnosis of learning, temporal analysis of agent behavior formation, and targeted interventions during training. We demonstrate these capabilities through extensive empirical studies spanning a range of RL environments and tasks.

**Experimental setup.**  We perform evaluation on a diverse suite of RL environments—navigation (`FrozenLake` and `MiniGrid`), classic control (`Acrobot` and `LunarLander`), driving (`Highway`), and locomotion (`BipedalWalker`)—covering discrete and continuous state and action spaces with varying complexity and reward structures. We defer descriptions of environments to Appendix A.1 and PPO training setups to Appendix A.2. Our code is at https://github.com/LDAORL/LDA-ORL.

## 4.1   Diagnosis of learning: what features bottom records?

In this section, we analyze the bottleneck that hinders learning in online RL. Specifically, we examine the bottom records for $f^{\text{return}}$ and uncover a consistent pattern across training rounds (additional examples in Appendix B.1): these bottom records are characterized by *inaccurate advantage estimates*, echoing observations in the literature [Ilyas et al., 2018].

Fig. 3(a–b) illustrates two examples. In `FrozenLake`, bottom records include poor actions receiving high positive $\hat{A}$ and good actions receiving negative $\hat{A}$. Similarly, in `MiniGrid`, the agent drifts from the goal but receives positive $\hat{A}$. These instances of *misleading* advantage estimates harm the learning.

---

[2]For GRPO [Shao et al., 2024], which uses a group-relative baseline rather than value-function baseline, the target function needs to be adjusted as well.

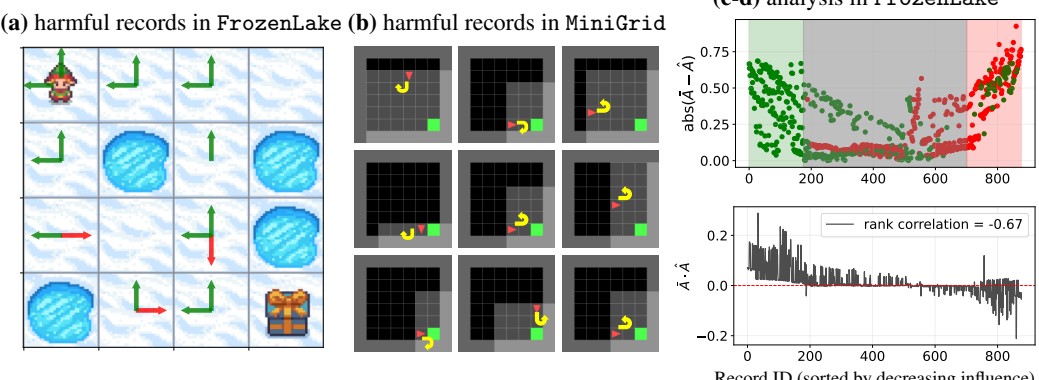

**(a)** harmful records in `FrozenLake` **(b)** harmful records in `MiniGrid`

**(c-d)** analysis in `FrozenLake`

Figure 3: **(a-b) Examples of bottom records**. (a) Bottom 100 records in `FrozenLake` at $k = 5$, aggregated over $(s, a)$ for demonstration: arrow indicates action, green/red for positive/negative $\hat{A}$. (b) Selected records among bottom 20 in `MiniGrid` at $k = 5$: ▼–agent, ■–goal, gray area–the limited egocentric observation, yellow arrows–agent action in $\{$turn left, turn right, forward$\}$; all records shown are of positive $\hat{A}$. **(c-d) These records are harmful due to their inaccurate advantage estimates**. We sort records by decreasing influence (top on the left). (c) $y$ axis is $|\bar{A} - \hat{A}|$; points with same/opposite signs for $\hat{A}$ and $\bar{A}$ colored green/red; top/bottom 20% region shaded green/red, and the intermediate in gray. (d) The product $\bar{A} \cdot \hat{A}$ versus record rank, showing a strong negative correlation.

We conduct quantitative analysis to characterize what constitutes "inaccurate" advantage estimates. We approximate the true advantage $A^\pi(s, a)$ using Monte Carlo (MC) rollouts from each $(s, a)$, averaging over multiple trajectories (details in Appendix B.4). We refer to this as the MC estimate, denoted by $\bar{A}$, and compare it with the advantage estimate $\hat{A}$. We perform analysis in `FrozenLake`.

Our analysis reveals two key aspects of "inaccuracy": (1) **Sign mismatch**: A significant proportion of bottom records exhibit opposite signs for the advantage estimate $\hat{A}$ and the MC estimate $\bar{A}$ (marked by red points in Fig. 3(c)). (2) **Large magnitude errors**: These records also have large $|\bar{A} - \hat{A}|$. Together, sign flips and large magnitude errors generate strong but misleading learning signals. Indeed, the Spearman rank correlation [Spearman, 1904] between each record's influence and the product $\bar{A} \cdot \hat{A}$ is strongly negative (Fig. 3(d)), confirming that misaligned advantages drive harmful gradient steps.

## 4.2 Temporal analysis of behavior formation: phase transition of top records

We investigate the reinforcement of a specific behavior ($a$ at $s$), characterized by a monotonic increase in $\pi(a|s)$. We track the evolution of top records w.r.t. $f^{\text{action}}$ across training rounds, which are critical in shaping the agent's behavior. Our analysis reveals an intriguing three-stage phase transition (Fig. 4).

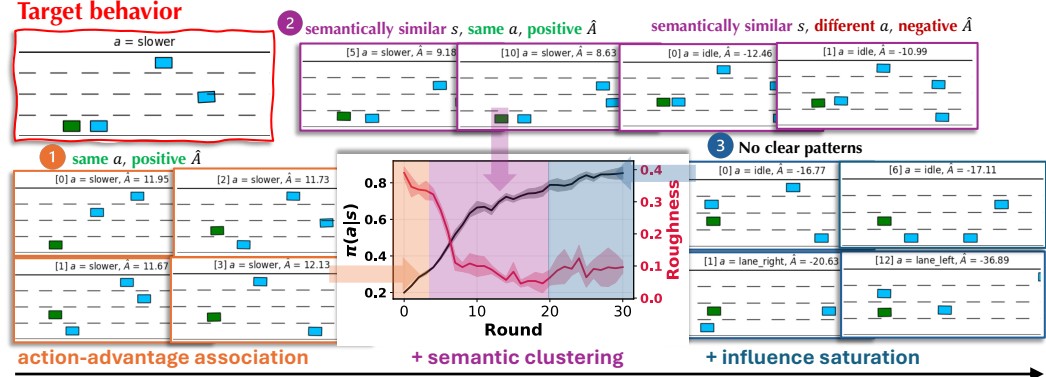

Figure 4: **Phase change of top records** in `Highway`, with the target behavior *taking the action "slower" when tailing the front vehicle*. In the inner plot, the black curve depicts $\pi(a|s)$; the red curve shows the measured roughness of the graph. ■: ego vehicle; ■: other vehicle. Three phases: ❶: simple action-advantage associations; ❷: semantic clustering (tailing states); ❸: no clear patterns.

1. **Initial association**: Initially, top records highlight patterns based on simple *action-advantage association*: they manifest target action paired with positive $\hat{A}$, or alternative actions paired with negative $\hat{A}$ (see Appendix B.2 for examples). The agent's behavior in this phase is reinforced through this naive association, largely ignoring the context of *state*. This basic association persists throughout training, even as more complex relationships are learned.

2. **Semantic clustering**: As learning progresses, the agent develops more nuanced representations. As a result, a pattern of *semantic clustering* develops alongside the initial action-advantage association. Top records in this phase demonstrate action-advantage association *within* states semantically similar to the target state, indicating the agent has learned to generalize across similar situations.

3. **Influence saturation**: In the final phase where learning approaches convergence, influence scores for most records stabilize near zero and become dominated by noise. Due to this noise, the top records appear less structured, though the action-advantage association still persists.

We quantify these phases by analyzing the *roughness* (normalized Dirichlet energy) [Von Luxburg, 2007] of a similarity graph, a measure closely related to the graph Laplacian [Chung, 1997]. In this graph, nodes represent records, values are ($L_\infty$-normalized) influence scores $\tilde{I}_i$, edge weights $w_{ij}$ capture semantic similarity and decay with embedding distance (details in Appendix B.2). Roughness, computed as $\sum w_{ij}(\tilde{I}_i - \tilde{I}_j)^2 / \sum w_{ij}$, is low when semantically similar records have similar influence; this captures the *clustering* effect. We track roughness across training rounds. As Fig. 4 shows, roughness remains high in Phase 1, indicating influence scores are largely uncorrelated with semantic similarity. It then significantly drops in Phase 2, representing the formation of semantically meaningful *clusters* of records with similar influences. In Phase 3, roughness remains low due to the settling of clustering, but exhibits minor fluctuations due to influence scores dominated by noise upon convergence.

## 4.3 Targeted interventions during training: filtering amplifies policy gain

Sec. 4.1 demonstrates that our framework can identify harmful training records, thereby opening possibilities for targeted interventions. As a sanity check, we apply a simple intervention procedure within *a single training round* to verify if removing these records yields performance gains.

Our procedure is straightforward: in round $k$, we identify records in $B^{(k)}$ with negative influence scores w.r.t. $f^{\text{return}}$, remove them, and re-train the agent on the filtered dataset starting from $\theta^{(k)}$. Fig. 5 shows that this consistently improves performance throughout learning and across environments.

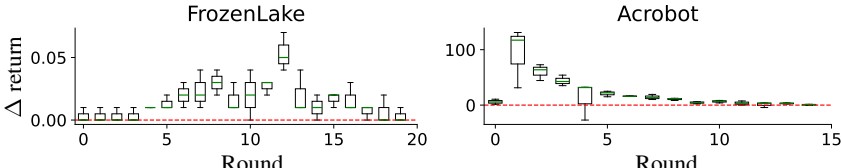

Figure 5: **Boxplots of $\Delta$ return for single round interventions in two environments**; red dashed line for zero $\Delta$. We intervene for each round *independently*. The $\Delta$ return is computed as the difference between the test return of the model trained on the *filtered* dataset and the *original* dataset. Results are shown for 3 random seeds. Additional results can be found in Appendix B.3.

A reader may ask: how can $f^{\text{return}}$ be meaningful when it relies on on-policy data with potentially inaccurate advantage estimates, unlike clean validation data used in traditional data attribution for supervised learning? Despite potential noise in individual records, the aggregated signal from $f^{\text{return}}$ is reasonably robust. This arises from the close alignment of $f^{\text{return}}$ with the PPO objective: effective PPO updates on the training buffer implies a reliable $f^{\text{return}}$ for attribution, enabling our intervention to clear away misleading records while retaining beneficial ones. This can be seen as *purifying* the learning signal, thereby *amplifying* the improvement achieved by PPO. More discussions are in Appendix B.3.

## 5 Iterative Influence-Based Filtering for Online RL Training

Standard online RL algorithms typically treat all collected experiences uniformly. However, as our analysis in Sec. 4.1 has shown, some records can be harmful for learning. This likely contributes to the notorious *sample inefficiency* of online RL, a challenge widely acknowledged [Yu, 2018]. Given this, a natural question arises: *can we leverage the local data attribution framework to tackle this challenge?*

We propose Iterative Influence-Based Filtering (IIF), building on the single-round interventions in Sec. 4.3. IIF filters records based on their computed influence scores, uses the resulting improved policy to sample new data, and repeats the cycle. This creates a loop for iterative refinement. We detail the algorithm below and showcase its effectiveness in traditional RL environments and RLHF for LLMs.

## 5.1 Algorithm and designs

---

**Algorithm 1:** Iterative Influence-Based Filtering (IIF) for Online RL

---

**Define:** $\mathcal{E}$: environment. $n$: # records in a rollout buffer. $p \in (0, 1]$: percentage of negative records to drop.

1 **Function** Update(model):
    ▷ Stage I: sampling
2     $B \leftarrow$ CollectTransitions($\mathcal{E}$, model, $n$)     ▷ collect transitions into buffer $B$
    ▷ Stage II: Filtering
3     $I \leftarrow$ ComputeInfluence(model, $B$)     ▷ compute influence for each record
4     $B_{\text{filtered}} \leftarrow$ DiscardBottomRecords($B, I, p$)     ▷ drop bottom records
    ▷ Stage III: training
5     **return** PPOUpdate(model, $B_{\text{filtered}}$)

6 **for** iter $= 1$ **to** $T$ **do**
7     model $\leftarrow$ Update(model)

---

Alg. 1 outlines IIF. Compared to standard PPO, IIF introduces an additional step of filtering (in red) between data collection and training. We further highlight the desiderata and IIF's design choices.

**Sample efficiency.** We aim to reduce the environment interactions required to reach a given performance level. To achieve this, IIF reuses the original rollout buffer $B^{(k)}$ as the validation set for influence calculation, incurring no extra sampling overhead. Furthermore, by selectively filtering bottom records, IIF accelerates learning, thus further reducing the total interactions needed.

**Computational cost.** We aim to keep the overhead of influence calculation small. This is achieved through two design choices. (1) Instead of iterating over all intermediate checkpoints, we compute the influence scores for the entire rollout buffer $B^{(k)}$ in round $k$ via $\left\langle \nabla_\theta f(\theta^{(k)}), \nabla_\theta \mathcal{L}^{\text{PPO}}(\theta^{(k)}, z_i) \right\rangle$, using only the initial parameter $\theta^{(k)}$. This saves a full training pass and excessive forward/backward calculations. (2) We implement an efficient "ghost dot product" following Wang et al. [2025a].

**Final performance.** We aim to improve the policy's final performance compared to standard training. IIF fulfills this through identifying and filtering out harmful records.

IIF employs a hyperparameter, $p$, which determines the amount of records to discard. We evaluate various $p$'s and report the best in Fig. 6. We observe that removing all negative-influence records ($p = 100\%$) as in Wang et al. [2025a] is often suboptimal, likely due to the non-additivity of sample influence [Hu et al., 2024]. Full ablation and recommendations for the choice of $p$ are in Appendix B.6.

## 5.2 Experiments in traditional RL environments

**Experimental setup.** We evaluate IIF on the diverse set of RL environments introduced in Sec. 4.

*Baselines*: We compare IIF with standard PPO and a random filtering baseline (dropping a similar fraction of records). We additionally investigate an advantage based filtering heuristic in Appendix B.4 motivated by the characterization of bottom records in Sec. 4.1, as well as a TD error based heuristic in Appendix B.5 inspired by the Prioritized Experience Replay algorithm [Schaul et al., 2016].

*Metrics*: We quantify sample efficiency by the reduction in training rounds required for IIF to match standard training. For a performance level $v$ (measured by test return), let $m_{\text{std}}(v)$ and $m_{\text{IIF}}(v)$ be the earliest training rounds where standard training and IIF achieve performance at least $v$, respectively. The reduction at $v$ is defined as $(1 - {m_{\text{IIF}}(v)}/{m_{\text{std}}(v)}) \times 100\%$. We report two metrics: $SE_{\text{ave}}$, the mean reduction over a list of strictly increasing performance levels reached by standard training, and $SE_{\text{peak}}$, the reduction at its peak. We measure computational cost by runtime; we similarly define $RT_{\text{peak}}$ as the reduction of runtime at the performance peak. Model performance is measured by the average test return over multiple episodes. See Appendix A.2 for further details on experimental setups.

**Results.** Fig. 6(a) presents the test returns for each environment; Fig. 6(b) summarizes the efficiency and runtime metrics. We report a detailed breakdown of runtime in Appendix B.9. Our key findings

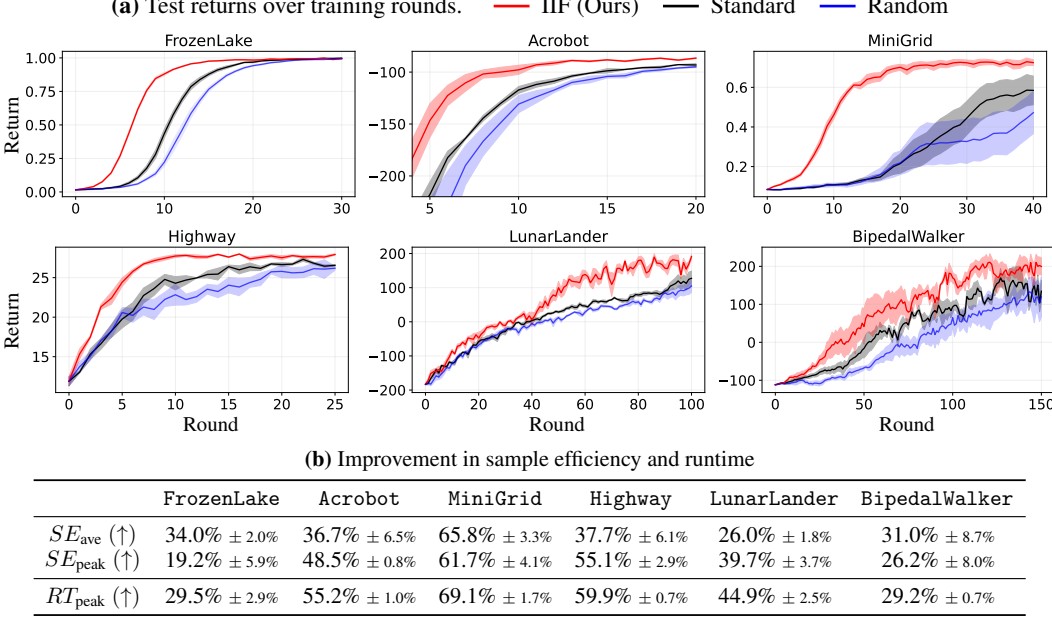

**(a)** Test returns over training rounds. — IIF (Ours) — Standard — Random

**(b)** Improvement in sample efficiency and runtime

|  | FrozenLake | Acrobot | MiniGrid | Highway | LunarLander | BipedalWalker |
|---|---|---|---|---|---|---|
| $SE_{\text{ave}}$ ($\uparrow$) | 34.0% $\pm$ 2.0% | 36.7% $\pm$ 6.5% | 65.8% $\pm$ 3.3% | 37.7% $\pm$ 6.1% | 26.0% $\pm$ 1.8% | 31.0% $\pm$ 8.7% |
| $SE_{\text{peak}}$ ($\uparrow$) | 19.2% $\pm$ 5.9% | 48.5% $\pm$ 0.8% | 61.7% $\pm$ 4.1% | 55.1% $\pm$ 2.9% | 39.7% $\pm$ 3.7% | 26.2% $\pm$ 8.0% |
| $RT_{\text{peak}}$ ($\uparrow$) | 29.5% $\pm$ 2.9% | 55.2% $\pm$ 1.0% | 69.1% $\pm$ 1.7% | 59.9% $\pm$ 0.7% | 44.9% $\pm$ 2.5% | 29.2% $\pm$ 0.7% |

Figure 6: (a) **Test returns over rounds for IIF vs. baselines.** IIF speeds up learning and improves performance. Results are averaged over 5 random seeds. For `Acrobot`, we omit early rounds where returns rise from -500 to -200 for better visualization. (b) **Sample efficiency and runtime metrics.**

are summarized as follows: 1) IIF achieves substantial sample efficiency gains, showing a 20-67% reduction in training rounds required to match the standard training performance across environments. 2) The computational overhead of IIF is negligible, and offset by the reduced optimization time (see Appendix B.9), leading to significant improvement in runtime. 3) IIF's final performance exceeds standard training in almost every environment. These observed gains stem from effective data attribution rather than mere data reduction: random filtering performs significantly worse than original training.

## 5.3 Extending IIF to RLHF for large language models

As the final part, we apply IIF to improve Reinforcement Learning from Human Feedback (RLHF).[3] Compared to standard PPO, RLHF introduces several key differences. First, the atomic unit shifts from state-action records to prompt-generation pairs,

where each generation is a *trajectory* (or sequence) of tokens. Second, RLHF incorporates *dual* reward sources: a reward model evaluating the final generation, and a per-token KL divergence penalty to constrain deviation from a reference model.

To accommodate these differences, we adapt IIF for RLHF by employing a sequence-level objective:

$$f^{\text{seq}}(\theta) = \mathbb{E}_{x \sim D_{\text{val}}, y \sim \pi^{\text{ref}}(\cdot|x)} \left[ \log \pi_\theta(y \mid x) \hat{A}^{\text{ref}}_{-1}(x, y) \right],$$

where $x$ is a prompt drawn from the validation set $D_{\text{val}}$, $y$ the generation, $\log \pi_\theta(y|x) = \sum_i \log \pi_\theta(y_i|x, y_0, \ldots, y_{i-1})$ the log-probability of the sequence $y$ given $x$, and $\hat{A}^{\text{ref}}_{-1}$ the advantage estimate at the last token. This objective emphasizes the reward model's feedback at the last token.

**Experimental results: toxicity mitigation.** We consider the task of detoxifying LLMs using RLHF [Hugging Face, 2023], using gpt-neo-2.7B [Black et al., 2021] as our base

**(a)** Training Reward ($\uparrow$)

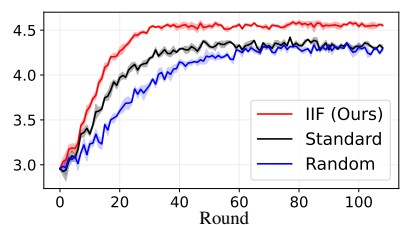

**(b)** Test toxicity ($\downarrow$) on a different test set, evaluated using a different toxicity detector.

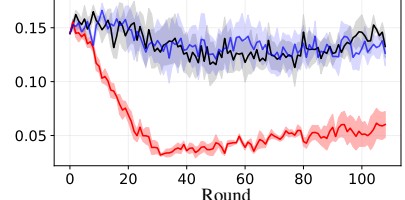

Figure 7: IIF improves the efficiency and performance of RLHF.

---

[3]Another line of work focuses on improving reward modeling in RLHF (the stage before PPO) via preference data selection [Muldrew et al., 2024, Das et al., 2024, Shen et al., 2025]; this is orthogonal to our work.

model. Fig. 7 illustrates the effectiveness of our approach. We defer detailed experimental setups to Appendix A.3 and additional results (e.g., comparisons with using the target function $f^{\text{return}}$) in Appendix B.11.

We further highlight IIF's substantial gains in *computational efficiency*. IIF filters out negative-influence records ($\sim$50% of all), effectively *halving* the optimization time per round. Furthermore, IIF accelerates learning, requiring less than *half* the number of rounds to surpass standard training, significantly enhancing sample efficiency. The overhead of influence calculation is minimal. Collectively, these factors result in an $\sim$4$\times$ reduction in total runtime (detailed breakdown in Appendix B.12).

## 6   Related Work

Interpretability in reinforcement learning has become a central research theme because real-world deployment requires agents that are trustworthy and reliable [Arulkumaran et al., 2017, Sutton and Barto, 2018, Milani et al., 2024, Cheng et al., 2025]. Early studies emphasize *feature*-level explanations: they highlight regions of the observation space that most influence an agent's decisions, often through saliency maps or attention heatmaps [Zahavy et al., 2016, Greydanus et al., 2018, Mott et al., 2019, Atrey et al., 2020, Puri et al., 2020]. A complementary thread seeks *policy*-level explanations. These works approximate learned policies with human-interpretable rules [Verma et al., 2018, Soares et al., 2020], design transparent architectures [Topin et al., 2021, Demircan et al., 2025], or dissect reward functions to clarify action choices [Juozapaitis et al., 2019, Liu and Zhu, 2025]. More recently, researchers have probed how entire training *trajectories* shape behavior [Deshmukh et al., 2023].

Zooming in further, identifying critical *states* offers a finer-grained view of decision making. Several approaches address offline settings [Guo et al., 2021, Yu et al., 2023, Liu et al., 2023, Rishav et al., 2025]. Closer to our focus are methods that target online RL such as lazy-MDP [Jacq et al., 2022], StateMask [Cheng et al., 2023] and RICE [Cheng et al., 2024]. Lazy-MDP augments the action space with a "lazy" action and penalizes non-lazy choices; states where the agent still acts are interpreted as important. However, this approach requires modifying the training pipeline. StateMask and RICE train an auxiliary mask network alongside the policy, forcing random actions in selected states while keeping returns roughly unchanged; masked states are deemed non-critical. Nevertheless, these methods crucially rely on the policy being sufficiently developed, which limits their applicability when agents are still learning in complex environments.

Moving beyond these constraints, our work introduces data attribution as a principled lens for interpretability in online RL. This approach closes a key methodological gap in the literature, delivers fresh insights for RL researchers and practitioners, and informs more efficient and effective training.

## 7   Conclusion and Limitations

This work pioneers data attribution for online RL by introducing a local attribution framework that addresses the circular dependency between data and model. The framework provides fine-grained insights into how training records shape model behaviors and offers a principled approach to enhancing the interpretability, efficiency, and effectiveness of online RL. We discuss a few limitations.

**Optimizers.**   Our framework leverages TracIn, which is designed for SGD [Hammoudeh and Lowd, 2024]. However, adaptive optimizers like Adam [Kingma and Ba, 2015] are prevalent in modern RL [Asadi et al., 2023] and LLMs [Zhao et al., 2025]. In this work, we follow Wang et al. [2025b] and employ SGD as a proxy for Adam. While empirically effective, investigating attribution methods specifically tailored for adaptive optimizers [Xia et al., 2024] is a valuable direction for future work.

**RL algorithms.**   Extending our framework to other online RL algorithms, particularly those used for LLMs like GRPO [Shao et al., 2024, DeepSeek-AI, 2025, Yu et al., 2025], is a promising avenue. Technically, our framework should generalize provided the attribution entity and per-sample gradients are well-defined. On the application side, leveraging attribution as a principled tool for improving LLM reasoning offers an intriguing alternative to existing data selection methods [Li et al., 2025, Shi et al., 2025, Xu et al., 2025, Wang et al., 2025c] that are largely based on heuristics.

**Counterfactual interpretation.**   Finally, our local attribution framework, while powerful, lacks a clear counterfactual interpretation. This limitation partly stems from TracIn itself, but primarily from the fundamental difficulty of tracking causal effects across the circular data-model dependency inherent in online RL, as discussed in Sec. 3. We encourage future work to tackle this open problem.

## Acknowledgements

We thank the anonymous NeurIPS 2025 reviewers for their constructive feedback. YH thanks Haozhe Si for assistance in setting up an NVIDIA instance. YH and JM thank Huazheng Wang and Kaiqing Zhang for helpful discussions on variance reduction. YH and HZ are partially supported by NSF IIS Grant No.2416897 and the NVIDIA Academic Grant Program. HZ also acknowledges support from a Google Research Scholar Award. Nan Jiang acknowledges funding support from NSF CNS-2112471, NSF CAREER IIS-2141781, Google Scholar Award, and Sloan Fellowship.

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

# A Detailed Experimental Setups

## A.1 Standard RL environments

We offer a detailed description of the RL environments used in our experiments in Table 1.

Gymnasium and `Highway` are licensed under MIT license; `MiniGrid` is licensed under Apache-2.0 license.

Table 1: **A summary description of RL environments we use in experiments.** Besides MiniGrid and Highway, other environments are from Gymnasium [Towers et al., 2024].

| Env | Env ID & Args | Goal | State Space | Action Space | Reward Structure |
|-----|---------------|------|-------------|--------------|------------------|
| `MiniGrid` [Chevalier-Boisvert et al., 2023] | `MiniGrid-Empty-8x8-v0`[4] | Navigate to a target location | $3 \times 7 \times 7$ image, representing the egocentric view of the agent's observation | 7 **discrete** actions: {turn left, turn right, move forward, pickup, drop, toggle, done} | **Sparse**: 1 - 0.9 (step_count/max_steps) on success, 0 otherwise |
| `FrozenLake` | `FrozenLake-v1`[5], map=4x4, slippery=False | Navigate from start to goal without falling into holes | 1 discrete integer: agent position index on the grid | 4 **discrete** actions: {Left, Down, Right, Up} | **Sparse**: +1 on reaching goal, 0 otherwise |
| `Acrobot` | `Acrobot-v1`[6] | Swing up the link to reach a target height | $\mathbb{R}^6$, providing information about the two rotational joint angles and their angular velocities | 3 **discrete** actions: $\{-1, 0, 1\}$ torque (N m) | **Dense**: -1 per step until reaching the target height |
| `Highway` [Leurent, 2018] | `highway-v0`[7], vehicle_count=10 | Drive at high speed while avoiding collisions | Kinematic Observation: $5 \times 5$ array of ego and nearby vehicles, including their location and speed | 5 **discrete** actions: {LANE_LEFT, IDLE, LANE_RIGHT, FASTER, SLOWER} | **Dense**: $(v-v_{\min})/(v_{\max}-v_{\min})-b\cdot$ collision at each step |
| `LunarLander` | `LunarLander-v2`[8] | Land safely on the pad from flight | $\mathbb{R}^8$: the coordinates of the lander, its linear velocities, angle, angular velocity, and whether each leg is in contact with the ground | 4 **discrete** actions: {do nothing, fire left, fire main, fire right} | **Dense**: +10 per leg contact; –0.03 per side-engine step; –0.3 per main-engine step; +100 on safe landing; –100 on crash; distance/velocity/angle terms |
| `BipedalWalker` | `BipedalWalker-v3`[9] | Traverse rough terrain without falling | $\mathbb{R}^{24}$: hull angle speed, angular velocity, horizontal & vertical speed, joints positions & angular speed, legs contact with ground, 10 lidar measurements | 4 **continuous** actions: motor speed values in $[-1, 1]$ for 4 joints at hips and knees | **Dense**: +1 per forward step; -100 on fall; small penalty proportional to torque magnitude |

## A.2 Experimental setups for standard RL

**Training setups.** We adopt `Stable-Baselines3`[10] [Raffin et al., 2021] (MIT license) as our training framework for the standard RL experiments. We use PPO [Schulman et al., 2017] as our RL algorithm and adopt the default training hyperparamters and network architectures for most environments unless otherwise specified.

---

[4] https://minigrid.farama.org/environments/minigrid/EmptyEnv/
[5] https://gymnasium.farama.org/environments/toy_text/frozen_lake/
[6] https://gymnasium.farama.org/environments/classic_control/acrobot/
[7] https://highway-env.farama.org/environments/highway/
[8] https://gymnasium.farama.org/environments/box2d/lunar_lander/
[9] https://gymnasium.farama.org/environments/box2d/bipedal_walker/
[10] https://stable-baselines3.readthedocs.io/en/master/index.html

- **Training hyperparameters:** We use n_steps=2048 (i.e., $n = |B^{(k)}| = 2048$), batch_size=64 (i.e., $|\mathcal{B}_j^{(k)}| = 64$), n_epochs=10 (i.e., each rollout buffer will be used for 10 epochs), learning_rate=5e-3 with optimizer=SGD in all environments except `BipedalWalker`, for which we use 3e-4 with Adam. total_timesteps per environment are: 102,400 for `FrozenLake` (50 rounds), 81,920 for `MiniGrid` (40 rounds), 102,400 for `Acrobot` (50 rounds), 204,800 for `Highway` (100 rounds), 307,200 for `LunarLander` (150 rounds), 1,024,000 for `BipedalWalker` (1000 rounds). Other hyperparameters include ent_coef=0.0, clip_range=0.2, gamma=0.99, gae_lambda=0.95, vf_coef=0.5, max_grad_norm=0.5.

- **Network architectures:** For `FrozenLake`, `Acrobot`, `Highway`, `LunarLander`, and `BipedalWalker`, we use the default `MlpPolicy` in Stable-Baselines3. This policy uses two-layer MLP networks (64 hidden units per layer), taking the flattened observation as input. For `MiniGrid` with image input, we use an adapted `CnnPolicy` with a custom feature extractor. The extractor comprises two convolutional layers (with 16 and 32 filters respectively, and 3x3 kernels) followed by a linear layer of 64 hidden units.

**Evaluation setups.** We evaluate the *stochastic* performance of each policy $\pi_{\theta^{(k)}}$ at every training round $k$ by averaging returns over multiple evaluation episodes. Specifically, we run 1000 episodes for `LunarLander`, `Acrobot`, `MiniGrid`, and `FrozenLake`; and 100 episodes for `Highway` and `BipedalWalker`.

## A.3 Experimental setups for RLHF

We follow Hugging Face [2023] to set up this experiment. The base model is a 2.7B parameter GPT-Neo model [Black et al., 2021] (MIT license).

**Training setups.** We adopt TRL[11] [von Werra et al., 2020] (Apache-2.0 license) as our training framework to fine-tune the based model via PPO. We employ LoRA [Hu et al., 2022] to perform PEFT fine-tuning, with a rank of 16, $\alpha$ of 32 and dropout of 0.05. The dataset for PPO training is `real-toxicity-prompts`[12] [Gehman et al., 2020] (Apache-2.0 license). For each example, we extract the first 10-15 tokens as a prompt, generate a 30-token continuation, and score it with the reward model, a toxicity detector `LFTW R4 Target`[13][Vidgen et al., 2021]. The reward signal is the raw logits of the label "neutral" of the detector.

The naming of the hyperparameters in TRL slightly differs from the ones in `Stable-Baselines3`. Here we stick to the naming in TRL to report the hyperparameters and clarfy their meanings using our notations. We follow Hugging Face [2023] to use batch_size=256 (i.e., $n = |B^{(k)}| = 256$), mini_batch_size=1 (i.e., $|\mathcal{B}_j^{(k)}| = 1$), ppo_epochs=4 (i.e., each rollout buffer will be used for 4 epochs), learning_rate=1e-5 with Adam optimizer, and all other default hyperparameters in TRL. We train for one epoch over the training dataset, which amounts to 109 rounds in total.

**Evaluation setups.** We evaluate the performance of each policy $\pi_{\theta^{(k)}}$ at every training round $k$. Evaluation is performed on `Wiki-Toxic`[14], which is of a different distribution than the training dataset. For each toxic sample, we use the full sample as the prompt (significanlty longer than used in training and thus more likely to elicit toxic continuations), and generate a 30-token continuation (same as the training setup). We then evaluate the toxicity of the generated continuation using another toxicity detector `da-electra-hatespeech-detection`[15]. Evaluation is conducted over 400 samples, and we report the mean toxicity probability.

---

[11]https://huggingface.co/docs/trl/index
[12]https://huggingface.co/datasets/allenai/real-toxicity-prompts
[13]https://huggingface.co/facebook/roberta-hate-speech-dynabench-r4-target
[14]https://huggingface.co/datasets/OxAISH-AL-LLM/wiki_toxic
[15]https://huggingface.co/alexandrainst/da-hatespeech-detection-base

# B  Additional Experimental Results

## B.1  More demonstrations of harmful records

**Harmful records for learning across training rounds.**   We examine the bottom records w.r.t $f^{\text{return}}$ in different training rounds $k$ and present the results in Fig. 8. (Results in the main paper, Fig. 3(a), corresponds to $k = 5$ here.)

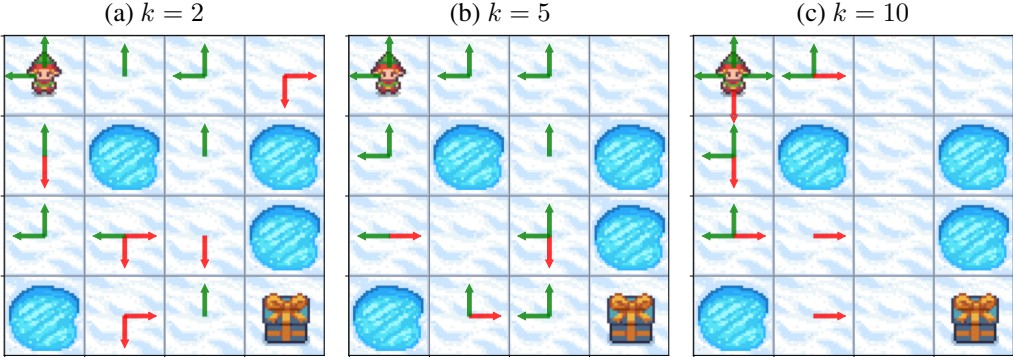

Figure 8: **Bottom records in different training rounds in** `FrozenLake`. Arrow indicates action, green/red indicates positive/negative $\hat{A}$.

Across all three snapshots ($k = 2, 5, 10$), the bottom records share a clear and consistent pattern: inaccurate advantage estimate, rewarding the agent for a poor action (moving away from the goal) and penalizing the agent for a good one (moving towards the goal).

**Harmful records in complex environments.**   We look into two complex environments. In `BipedalWalker` (locomotion), our analysis reveals bottom records where the agent was incorrectly penalized with a large negative advantage for executing a successful recovery move (e.g., applying corrective torque with a deeply bent knee ($\sim$35°) during landing or push-off). (We omit the visualizations for this environment as it does not conveniently support rendering given status vectors; the above analysis is done based on direct analysis of values in status vectors.) In `Pong` (Atari), we find that bottom records filtered by IIF consist of uninformative transitions (the ball being out of play or already moving away from the agent) that receive (inaccurately) high advantage estimates. By filtering out these samples, IIF achieves significant improvement in training efficiency. These results show that 1) bottom records feature inaccurate advantage estimates; 2) IIF is effective, holding generally across different environments. Examples are shown in Fig. 9.

## B.2  Quantifying phase change via weighted graph roughness analysis

**Measurement protocol.**   We provide full details of our quantitative investigation.

For each round $k$, we build the similarity graph $\mathcal{G}_k$ using records with positive influence scores in $B^{(k)}$ and their influence scores [Von Luxburg, 2007]. We embed each record $z_i$ as a node in the graph, with the node value being the $L_\infty$-normalized influence score $\tilde{I}_i = I_i / \|I\|_\infty$, the node embedding being the record embedding $e_i$ extracted by a well-trained network (obtained at the end of the PPO training). We set edge weights by a Gaussian kernel $w_{ij} = \exp(-\|e_i - e_j\|^2 / \sigma^2)$ with $\sigma$ chosen via the median-distance heuristic. We retain each node's $u$ nearest neighbors when building the similarity graph. This reduces computational cost. In practice, we find that varying $u$ from 20 to 100 has little effect on the roughness measure.

With the graph $\mathcal{G}_k$ built, we compute the graph roughness as follows:

$$\text{Roughness}(\mathcal{G}_k) = \frac{\sum_{i<j} w_{ij}(\tilde{I}_i - \tilde{I}_j)^2}{\sum_{i<j} w_{ij}}$$

We repeat this process for all rounds $k$ and plot the change of roughness over rounds.

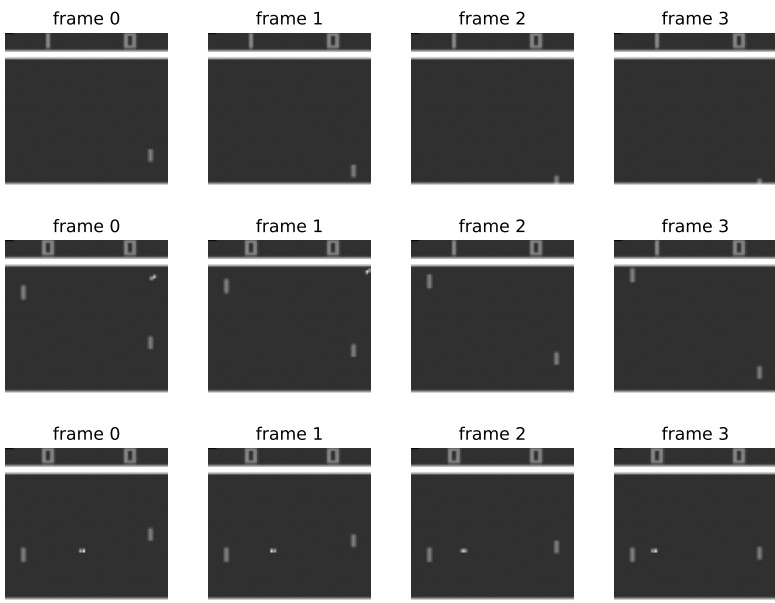

Figure 9: **Bottom records for the Pong.** The top and middle figures correspond to the case where the ball it out of play. The bottom figure corresponds to the case where the ball is moving away from the agent. (Note that in Pong, the ego agent is the one on the right.)

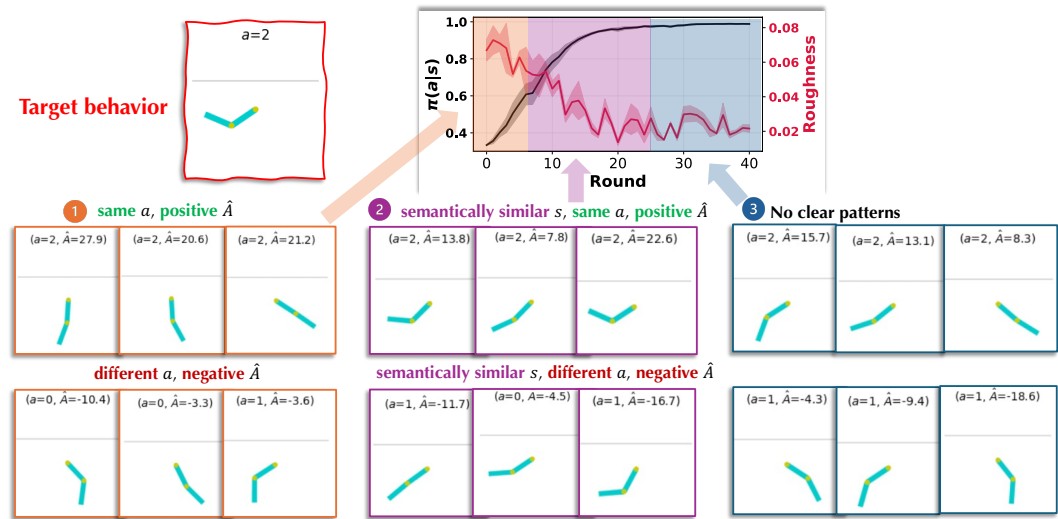

Figure 10: **Phase change of top records in** `Acrobot`.

**Results in more environments.** We study another environment `Acrobot`, investigating the phase change and measuring the roughness metric across rounds. The results are presented in Fig. 10. We observe a consistent trend of the three phases, aligned with the findings discussed in Sec. 4.2.

In Phase 1, top records include those with the same action and positive $\hat{A}$, and those with alternative actions and negative $\hat{A}$. Roughness is high in this phase. In Phase 2, semantically similar records (that consistently show the action-advantage association) emerge as top records; roughness decreases significantly in this phase. In Phase 3, learning approaches convergence and the semantic clustering stabilizes; influence scores become dominated by noise, causing roughness to show minor fluctuations.

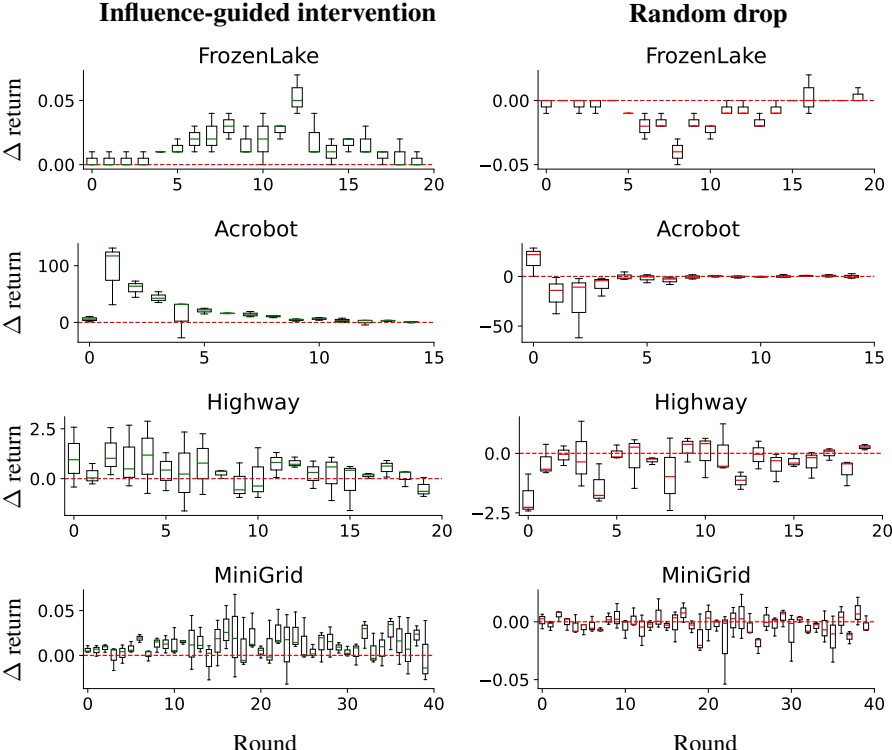

Figure 11: **Boxplots of $\Delta$ return for single rollout interventions in four environments, comparing influence-guided intervention (left) with random drop (right)**. We perform intervention for each iteration *independently* by removing bottom records and then retrain the model. The $\Delta$ return is calculated as the difference between the return from the model trained on the *filtered* dataset and the *original* dataset. Results are shown for three random seeds.

## B.3 Additional results for single-round intervention

Fig. 11 (as an extension of Fig. 5) presents the results of single-round interventions in four environments, additionally comparing with the random baseline that discards a similar amount of records.

We discuss several key takeaways: (1) Influence-guided intervention mostly leads to performance gains, while random drop mostly leads to performance degradation. (2) When standard PPO fails to improve (e.g. a dip at round $k = 9$ in Highway; see Fig. 6), the attribution signal can become unreliable, producing negative $\Delta$ return (see Fig. 11 at $k = 9$ in Highway), leading occasionally to interventions that fail to bring any improvement. However, as long as PPO's overall trend is upward, our intervention can effectively *purify* the learning and and drive net improvement over the full run.

We also note that while our approach has a flavor of *variance reduction*, in the sense that it removes outlier gradients, it is fundamentally different from standard variance reduction techniques such as Generalized Advantage Estimation [Schulman et al., 2016] or baseline extraction [Schulman et al., 2017, Sutton and Barto, 2018]. In particular, the analysis in Sec. 4.1 shows that our method identifies genuinely *harmful* rather than *useless* samples, and thus has a bias-correction effect.

## B.4 Advantage-based heuristic

**Method.** Sec. 4.1 characterizes the properties of the bottom harmful records—*sign mismatch* and *large magnitude errors*. Inspired by these findings, we design the following two heuristics for experience filtering:

- Heuristic 1: We discard records with opposite signs for $\bar{A}$ and $\hat{A}$. Among these records, we sort them by $|\bar{A} - \hat{A}|$ and discard the top $p\%$ records with the largest error.

- Heuristic 2: We discard records with opposite signs for $\bar{A}$ and $\hat{A}$. Among these records, we sort them by $\bar{A} \cdot \hat{A}$ and discard the bottom $p\%$ records with the smallest product (i.e., the most negative).

**Implementation.** These heuristics fundamentally rely on obtaining a reliable estimate of the true advantage function, $\bar{A}^\pi(s,a)$, for each training record. We obtain $\bar{A}$ using Monte Carlo (MC) estimates, i.e.,

$$\bar{A}^\pi(s,a) = \bar{Q}^\pi(s,a) - \bar{V}^\pi(s) = \mathbb{E}\left[\sum_k \gamma^k r_{t+k} | s_t = s, a_t = a\right] - \mathbb{E}\left[\sum_k \gamma^k r_{t+k} | s_t = s\right],$$

In environments with small, discrete state and action spaces, we can leverage the collected rollout buffer $B^{(k)}$ to obtain the estimate $\bar{A}^{\pi_{\theta^{(k)}}}$, as $B^{(k)}$ itself would include multiple occurrences of $(s,a)$ pairs or visits to state $s$, allowing for empirical averaging.

However, in environments with large discrete or contiuous state/action spaces, specific state-action pairs $(s,a)$ are rarely encountered multiple times in $B^{(k)}$. Accurately estimating $\bar{A}^{\pi_{\theta^{(k)}}}(s,a)$ for each record in these more complex settings would require resetting the environment to the specific $s$ and then performing numerous independent rollouts under policy $\pi_{\theta^{(k)}}$. This procedure is generally computationally infeasible.

For consideration of computational efficiency, in our study below, we limit to environments with *discrete* state and action spaces, where we compute $\bar{A}$ using the collected rollout buffer $B^{(k)}$, instead of performing additional sampling in the environment.

**Results.** Fig. 12 compares the two advantage-based heuristics against IIF and standard training in `FrozenLake` and `MiniGrid`.

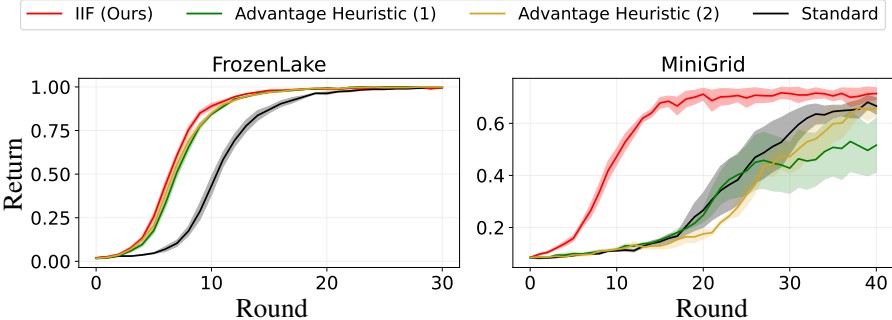

Figure 12: **Test returns over training rounds for the two advantage-based heuristics**, compared with IIF and standard PPO. Results are averaged over three random seeds.

In `FrozenLake`, a small discrete environment, both heuristics closely match IIF's learning curve and final return, and substantially outperforms standard PPO. This result serves as a validation of our initial findings in Section 4.1, confirming that transitions exhibiting sign mismatch or large advantage estimation errors are indeed key properties of harmful experiences, and that filtering based on these properties can significantly improve training efficiency.

However, in `MiniGrid`, which features a significantly larger state space, the advantage-based heuristics fail to improve upon the standard PPO baseline and in fact even degrade performance. There are two possible reasons. (1) The advantage estimates $\bar{A}$ are noisy due to the limited number of repeated visits per $(s,a)$ and $s$ in $B^{(k)}$, leading to inaccurate filtering. (2) These heuristics rely solely on the relationship between estimated and true advantages; in comparison, IIF's influence score, derived from gradients, captures a broader, more nuanced set of characteristics of harmful records. This richer representation allows IIF to perform effective filtering when simple advantage heuristics fail.

In summary, these results validate our core insights: properties like sign mismatch and large estimation errors are indeed indicative of harmful training records. At the same time, their failure in more complex environments highlights the limitations of these simple heuristics. Our IIF framework,

by contrast, is more generally applicable; its influence scores capture a broader and more nuanced understanding of records' values beyond simple advantage discrepancies, enabling effective filtering even in complex domains.

### B.5   TD error based heuristic

**Motivation.**   Prioritized Experience Replay (PER) [Schaul et al., 2016] demonstrate that reweighting transitions in proportion to their temporal-difference (TD) error accelerates learning and improves performance in **off-policy** methods. TD error serves as a useful heuristic, indicating how "surprising" or "important" a transition is for updating the *value function*. While PPO is an on-policy method that typically uses a smaller, on-policy rollout buffer rather than a large replay buffer like those in off-policy algorithms, the core idea of focusing learning on more impactful experiences remains relevant. Inspired by PER, we investigate integrating a TD error based reweighting mechanism into the PPO training process to prioritize samples within its rollout buffer.

**Implementation.**   For each transition $(s_i, a_i, r_i, s'_i)$ collected and stored in the rollout buffer $B^{(k)}$, we first compute its TD error. The TD error for record $i$ is defined as:
$$\delta_i = r_i + \gamma V^{\pi_{\theta^{(k)}}}(s'_i) - V^{\pi_{\theta^{(k)}}}(s_i),$$
where $V^{\pi_{\theta^{(k)}}}$ denotes the current value function estimate (under the current policy $\pi_{\theta^{(k)}}$).

We then assign a priority to each record using a rank-based approach following Schaul et al. [2016]. We sort all transitions in the buffer $B^{(k)}$ in descending order based on the absolute value of their TD error, $|\delta_i|$. The base priority for transition $i$ is set as $P_i = 1/\text{rank}(i)$, where $\text{rank}(i)$ denotes the rank of transition $i$. Then, the probability of sampling record $i$ is
$$w_i = \frac{P_i^\alpha}{\sum_{j \in B^{(k)}} P_j^\alpha}, \quad \text{where } \alpha = 0.6 \text{ (following Schaul et al. [2016])}$$
This weighting scheme ensures that transitions with larger absolute TD errors receive higher emphasis during the PPO optimization steps.

**Results.**   We evaluate the performance of the TD error based reweighting heuristic by comparing it against our IIF and standard PPO on `FrozenLake` and `LunarLander`. Fig. 13 presents the test returns over training rounds for these approaches.

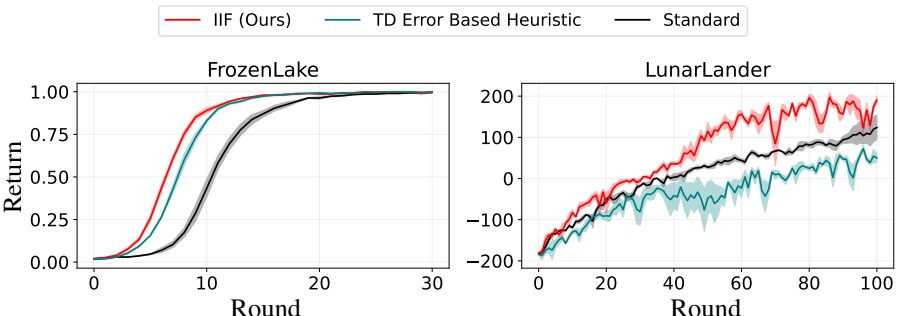

Figure 13: **Test returns over training rounds for the TD error based heuristic**, compared with IIF and standard PPO. Results are averaged over three random seeds.

In `FrozenLake`, a simple environment, both TD error and IIF accelerate convergence, reaching optimal return sooner. The TD error heuristic nearly matches IIF's speed, confirming that large TD errors align well with truly *useful* transitions when the state-action space is small and reward structure simple.

In contrast, in the more complex `LunarLander`, the TD error heuristic degrades performance: it learns more slowly than even standard PPO and exhibits greater variance. Although this heuristic succeeds in PER, we comment that there are intrinsic differences in the off-policy scenario where PER was proposed and evaluated, vs. the on-policy scenario (e.g., PPO) we study in this paper (Fig. 1). PER applies the TD error heuristic on a vast, diverse buffer. However, in PPO, raw TD errors mix estimator noise with true signal; PPO's small, fresh, on-policy batches exacerbate that noise; Our influence scores, in comparison, appears more robust in such scenarios.

## B.6 IIF performance under various filtering percentages

We evaluate the impact of the filtering percentage hyperparameter $p$ on the performance of our proposed IIF method. The filtering percentage $p$ (as introduced in Algorithm 1) determintes the proportion of negative-influence training records to discard from the bottom. We explore a wide range of values for $p \in \{100.0\%, 50.0\%, 25.0\%, 12.5\%, 6.25\%\}$, reducing the percentage by half at each level. Note that $p = 100.0\%$ means discarding all negative-influence records.

Fig. 14 shows the test returns over training rounds for IIF with varying $p$'s compared to baselines. We additionally quantify their efficiency using two metrics: $SE_{\text{ave}}$ and $SE_{\text{peak}}$ (introduced in Sec. 5.2). We summarize these efficiency statistics in Table 2.

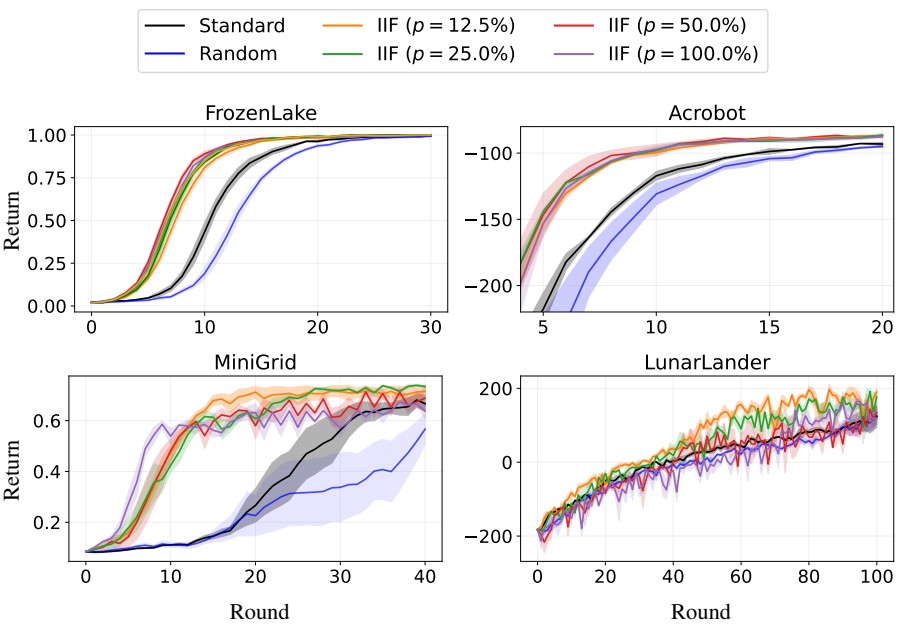

Figure 14: **Test returns over training rounds for IIF with a range of filtering percentages $p$**, compared to the baselines. Larger $p$ means more aggressive filtering. Results are averaged over three random seeds.

Table 2: **Sample efficiency comparison across varying filtering percentages.** Results show the improvement in sample efficiency metrics ($SE_{\text{ave}}$ and $SE_{\text{peak}}$) for different filtering percentages, across simpler and more complex environments. **Bold** values indicate the best performing value of $p$; *italicized* values show the second best. Results are averaged over three runs.

(a) $SE_{\text{ave}}$ ($\uparrow$)

|  | FrozenLake | Acrobot | MiniGrid | LunarLander |
|---|---|---|---|---|
| $p = 12.5\%$ | $23.5\% \pm 3.1\%$ | $29.2\% \pm 0.8\%$ | $67.5\% \pm 5.1\%$ | $\mathbf{28.2\%} \pm 1.3\%$ |
| $p = 25.0\%$ | $30.5\% \pm 3.3\%$ | $35.1\% \pm 0.6\%$ | $60.3\% \pm 10.6\%$ | $22.7\% \pm 5.6\%$ |
| $p = 50.0\%$ | $\mathbf{33.7\%} \pm 3.4\%$ | $\mathbf{36.7\%} \pm 6.5\%$ | $67.0\% \pm 5.3\%$ | $10.2\% \pm 6.5\%$ |
| $p = 100.0\%$ | $32.7\% \pm 1.7\%$ | $35.0\% \pm 0.5\%$ | $\mathbf{75.4\%} \pm 3.6\%$ | $8.9\% \pm 2.0\%$ |

(b) $SE_{\text{peak}}$ ($\uparrow$)

|  | FrozenLake | Acrobot | MiniGrid | LunarLander |
|---|---|---|---|---|
| $p = 12.5\%$ | $15.6\% \pm 5.1\%$ | $31.5\% \pm 2.2\%$ | $\mathbf{67.4\%} \pm 4.4\%$ | $\mathbf{41.6\%} \pm 5.7\%$ |
| $p = 25.0\%$ | $\mathbf{22.1\%} \pm 7.4\%$ | $\mathbf{48.5\%} \pm 0.8\%$ | $58.8\% \pm 13.1\%$ | $32.9\% \pm 13.1\%$ |
| $p = 50.0\%$ | $19.6\% \pm 8.4\%$ | $\mathbf{48.5\%} \pm 0.8\%$ | $50.6\% \pm 20.7\%$ | $15.5\% \pm 17.1\%$ |
| $p = 100.0\%$ | $15.9\% \pm 5.5\%$ | $43.1\% \pm 5.7\%$ | $54.9\% \pm 22.5\%$ | $15.8\% \pm 7.3\%$ |

We highlight several key findings:

- **Discarding all negative records ($p = 100\%$) is suboptimal.** As shown in Figure 14, setting $p = 100\%$ leads to suboptimal final performance, slower learning progress (also reflected in Table 2), and instability in training. This observation aligns with the concept of non-additivity of sample influence [Hu et al., 2024].

- **Any level of filtering improves performance over standard training.** Applying IIF with almost any filtering percentage demonstrates improvement compared to standard training. This underscores the general effectiveness of IIF in mitigating negative influence by removing a portion of identified negative samples.

- **The optimal filtering percentage varies with environment complexity.** In simpler environments (e.g. `FrozenLake`, `Acrobot`), removing half of the negative samples ($p = 50\%$) yields the best performance overall—simple environments could involve plenty of redundancy; aggressive pruning focuses learning on the most informative transitions. In contrast, in more complex environments (`MiniGrid`, `LunarLander`), the interplay among records is subtler: overly large filtering discard borderline-useful transitions, while a gentler filtering ($p = 12.5\%$) can achieve better performance.

Based on these findings, for our main experiments (see Sec. 5.2) we choose the specific filtering percentages to reflect the optimal configuration per environment. We use $p = 50\%$ for `FrozenLake`, `Acrobot`, `Highway`; $p = 12.5\%$ for `MiniGrid`, `LunarLander`; and $p = 6.25\%$ for `BipedalWalker`.

## B.7 Evaluating IIF with the Adam optimizer

Our main experiments in traditional RL environments are conducted using the SGD optimizer (see Appendix A.2). Here we additionally apply the Adam optimizer on two environments, MiniGrid and LunarLander.

We report the test return in Fig. 15, and sample efficiency and runtime metrics in Table 3. One observation is that IIF gains less with Adam compared to SGD in MiniGrid, whereas the trend is reversed for LunarLander (see Fig. 6 for reference). This is partly because Adam significantly speeds up training compared to SGD in MiniGrid (and thus reduces the room of improvement), but less so in LunarLander.

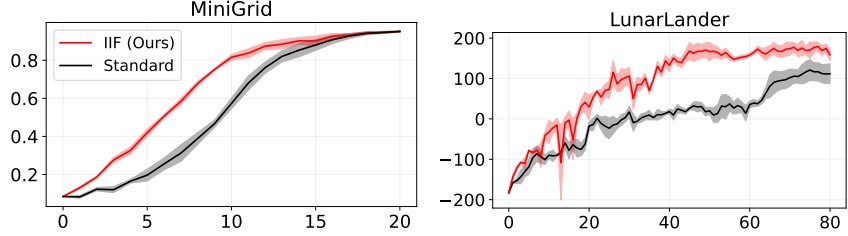

Figure 15: **Test returns over rounds for IIF vs. the standard training baseline, when using the Adam optimizer.** Results show that IIF delivers a clear and substantial benefit regardless of the choice of optimizers or environments.

Table 3: **Sample efficiency and runtime comparisons when using the Adam optimizer.**

|  | MiniGrid | LunarLander |
|---|---|---|
| $SE_{\text{ave}}$ ($\uparrow$) | 24.1% $\pm 1.4\%$ | 46.7% $\pm 4.5\%$ |
| $SE_{\text{peak}}$ ($\uparrow$) | 13.3% $\pm 3.1\%$ | 62.2% $\pm 5.0\%$ |
| $RT_{\text{peak}}$ ($\uparrow$) | 18.5% $\pm 1.0\%$ | 65.9% $\pm 3.2\%$ |

## B.8 Statistical significance of final performance gains

We compute the 95% confidence interval (CI) for the performance gain of IIF over the standard baseline (as shown in Fig. 6(a)). Concretely, we compute half-width = $t_{0.957,4} \times SE = 2.776 \times SE$. Results in Table 4 confirm a statistically significant improvement in the performance gain.

Table 4: **95% confidence interval (CI) for the performance gain** of IIF over the standard baseline across 5 random seeds.

|  | MiniGrid | LunarLander | BipedalWalker |
|---|---|---|---|
| 95% CI | [0.04, 0.33] | [22.54, 130.52] | [24.40, 75.99] |

## B.9 Runtime for experiments on traditional RL environments

We report the runtime for experiments on traditional RL environments in Table 5.

For **per-round runtime**, we report the time for the influence calculation step and the optimization step. The overhead of IIF in the influence calculation step is negligible. As IIF discards $p\%$ of the negative records, it enjoys a reduction in optimization time.

For **total runtime**, we first report the runtime for all training rounds (labeled as "All rounds"), and then report the runtime corresponding to the (reduced) rounds needed for IIF to match the peak performance of standard PPO (labeled as "Matching peak"). IIF's improvement in sample efficiency leads to a further speedup.

Finally, we report $RT_{\text{peak}}$ (also presented in Fig. 6(b)), calculated as the reduced percentage of wall clock time for IIF to match standard PPO. In summary, IIF presents a 29%-67% reduction in runtime, effectively speeding up learning.

Table 5: **Per-round runtime and total runtime (in seconds), as well as the percentage of overall reduced runtime for experiments on traditional RL environments.** Results are averaged over 3 training runs each for IIF and standard training. A dash (—) indicates that a measure is not applicable.

|  |  | FrozenLake | | Acrobot | | MiniGrid | |
|---|---|---|---|---|---|---|---|
|  |  | IIF | standard | IIF | standard | IIF | standard |
| **Per-round runtime** | Influence calc | $0.11_{\pm 0.01}$ | — | $0.25_{\pm 0.01}$ | — | $0.25_{\pm 0.02}$ | — |
|  | Optimization | $1.51_{\pm 0.04}$ | $2.01_{\pm 0.05}$ | $1.42_{\pm 0.02}$ | $2.02_{\pm 0.02}$ | $4.52_{\pm 0.06}$ | $5.02_{\pm 0.07}$ |
| **Total runtime** | All rounds | $82.15_{\pm 2.93}$ | $93.85_{\pm 2.68}$ | $70.01_{\pm 0.72}$ | $79.87_{\pm 1.00}$ | $365.23_{\pm 3.11}$ | $378.41_{\pm 2.98}$ |
|  | Matching peak | $64.64_{\pm 3.98}$ | — | $35.80_{\pm 0.79}$ | — | $107.43_{\pm 3.32}$ | — |
| $RT_{\text{peak}}$ **(reduced runtime %) (↑)** |  | $31.27\%_{\pm 3.28\%}$ | | $55.16\%_{\pm 1.04\%}$ | | $71.59\%_{\pm 1.05\%}$ | |

|  |  | Highway | | LunarLander | | BipedalWalker | |
|---|---|---|---|---|---|---|---|
|  |  | IIF | standard | IIF | standard | IIF | standard |
| **Per-round runtime** | Influence calc | $0.13_{\pm 0.02}$ | — | $0.13_{\pm 0.01}$ | — | $0.12_{\pm 0.01}$ | — |
|  | Optimization | $2.39_{\pm 0.48}$ | $3.29_{\pm 0.59}$ | $1.85_{\pm 0.04}$ | $2.05_{\pm 0.01}$ | $3.09_{\pm 0.20}$ | $3.30_{\pm 0.23}$ |
| **Total runtime** | All rounds | $214.41_{\pm 0.22}$ | $233.66_{\pm 0.24}$ | $318.68_{\pm 1.27}$ | $328.79_{\pm 3.65}$ | $676.78_{\pm 4.71}$ | $691.28_{\pm 13.33}$ |
|  | Matching peak | $93.73_{\pm 1.69}$ | — | $183.64_{\pm 6.69}$ | — | $489.55_{\pm 4.71}$ | — |
| $RT_{\text{peak}}$ **(reduced runtime %) (↑)** |  | $59.89\%_{\pm 0.72\%}$ | | $44.11\%_{\pm 2.29\%}$ | | $29.16\%_{\pm 0.66\%}$ | |

## B.10 Difficulty based heuristic

Inspired by the difficulty-based filtering (e.g., pass@k) primarily used to improve LLM Reasoning (RLVR) in GRPO [Yu et al., 2025, Bae et al., 2025], we develop a difficulty-based filtering approach for PPO. Concretely, we use reward as a proxy for difficulty and filter records receiving top and bottom rewards. However, this heuristic performs worse than random because it systematically removes data with both highest and lowest influence scores, thereby harming the learning process. This finding aligns with our results in Appendix B.5 for traditional RL, where an analogous heuristic using TD error as a proxy for difficulty also proved ineffective. Therefore, our evidence shows that while valid for GRPO, difficulty-based filtering is an ineffective heuristic for PPO.

## B.11 Comparing two target functions for RLHF

In the main text (Sec. 5.3), we introduced two target functions for RLHF: the standard one $f^{\text{return}}$, and an adapted sequence-level objective $f^{\text{seq}}$. Here we show the comparison of the two in Fig. 16.

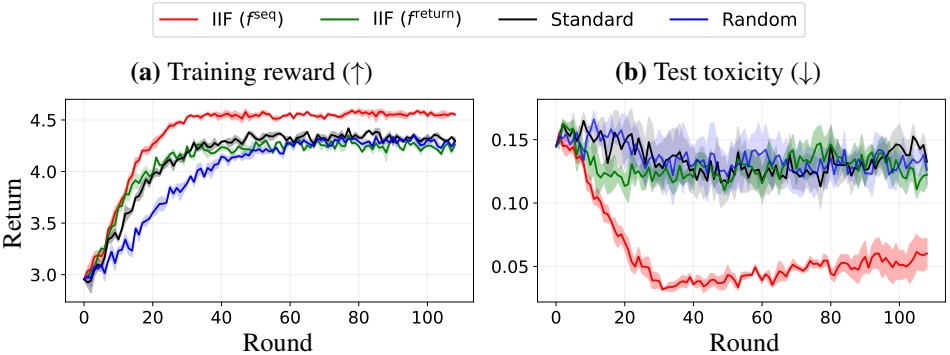

Figure 16: **Comparing two target functions $f^{\text{seq}}$ with $f^{\text{return}}$ for RLHF**. Results are averaged over 3 random seeds.

Overall, from both the training and testing curves, IIF with $f^{\text{seq}}$ clearly outperforms the others. Although IIF with $f^{\text{return}}$ initiallly improves faster than standard PPO, it soon plateaus, eventually converging to the same levels as the standard PPO baseline. This highlights that, the adapted sequence-level objective is more effective in RLHF's trajectory-centric setting with dual reward signals.

## B.12 A breakdown of runtime for the RLHF experiments

Table 6 breaks down the wall-clock time (in seconds) for each component of one RLHF training round, under standard PPO and our IIF. The overhead of influence calculation in IIF is significantly offset by reduced optimization time, leading to a $2\times$ speedup *per round*.

Beyond this per-round saving, IIF requires fewer rounds to achieve comparable performance with standard PPO (requiring $32.75\% \pm 1.52\%$ of training rounds, taking up $16.82\% \pm 1.32\%$ of runtime combined with per-round speedup). Furthermore, IIF reaches convergence to a higher reward faster as well (requiring $48.51\% \pm 2.44\%$ of training rounds, taking up $24.90\% \pm 0.80\%$ of wall-clock time). This marks a $4\times$ overall speedup plus performance improvement compared to standard PPO.

Table 6: **Per-round runtime (in seconds) for RLHF with IIF vs. standard PPO.** IIF halves optimization time by pruning $\sim 50\%$ of the data each round, while the overhead of influence calculation is negligible. Reported results are averaged over all 109 training rounds in 3 training runs (using 3 random seeds). A dash (—) indicates that a measure is not applicable.

|  | IIF | Standard PPO | % |
|---|---|---|---|
| Response generation & scoring | $1.71 \pm 0.06$ | $1.59 \pm 0.05$ |  |
| Forward | $1.03 \pm 0.04$ | $0.99 \pm 0.00$ |  |
| Influence calculation | $2.15 \pm 0.02$ | — |  |
| Optimization | $40.39 \pm 0.35$ | $85.56 \pm 0.17$ |  |
| **Total per-round runtime** | $45.28 \pm 0.47$ | $88.15 \pm 0.22$ | 51.37% |

## C Compute resources

All experiments were conducted on two Linux servers:

- **Machine 1**: Dual Intel Xeon Silver 4314 CPUs (16 cores/socket, 64 threads total), 251 GiB RAM, 4 NVIDIA RTX A6000 GPUs (48 GiB VRAM each).
- **Machine 2**: Dual AMD EPYC 7J13 CPUs (64 cores/socket, 256 threads total), 2 TiB RAM, 4 NVIDIA A100-SXM4-80GB GPUs (80 GiB VRAM each).

For experiments on standard RL benchmarks, we use both Machine 1 and 2; for experiments on RLHF, we use Machine 2 only.

All runtime results reported in Appendix B.9 were measured on Machine 1; all runtime results in Appendix B.12 were measured on Machine 2.

