# OpenReview forum: "A Snapshot of Influence: A Local Data Attribution Framework for Online Reinforcement Learning"
_NeurIPS.cc/2025/Conference — NeurIPS 2025 oral_

### Official Review · Reviewer_1ZdL · 2025-06-30

**Clarity:** 2
**Significance:** 3
**Originality:** 3
**Rating:** 5
**Confidence:** 3

**Summary:**

The paper adapts a data attribution framework (TracIn) for supervised learning for the on-policy RL algorithm PPO. The authors motivate why traditional attribution methods would fail in the RL setting, namely, because they are designed for static datasets. They adapt TracIn to on-policy RL by i) treating the rollouts collected for one update epoch as a “local” dataset, and (ii) adapting TracIn-style gradient similarity for RL by using action log-probabilities and advantage-weighted action log-probabilities. The paper highlights three use cases for data attribution in RL: diagnosing learning pathologies (wrong advantage estimates), identifying temporal phases of learning, and single-round data filtering. Building on the last use case, the authors propose **Iterative Influence-based Filtering (IIF)**: After every round they drop some records with negative influence and train PPO on a filtered buffer. The experiments show improvements in sample efficiency and wall clock time across a range of RL tasks (discrete/continuous control, dense/sparse rewards), as well as improved toxicity mitigation in RLHF.

**Questions:**

- Can the authors comment on possible extensions to off-policy algorithms? What could an extension look like?
- I'm a bit confused about the TracIn calculation, e.g., here
$\eta_j \sum_{i \in \mathcal B_j} \nabla_\theta f(\theta_j) \cdot \nabla_\theta \ell(\theta_j, z_i)~.$
Why does the first multiplication term not depend on a transition $z_i$?
- Is the baseline in Figure 6 using Adam? If not, how does your method compare to PPO+Adam? I think this is relevant because it is a limitation of your method and not PPO in general that SGD has to be used.
- It seems that IIF is most helpful in sparse-reward settings (Minigrid). Can you comment on the mechanism of why this is the case?
- $f^\text{action}$ is only used in Section 4.2. Could it also be used to speed up training? If so, how?

**Ethical Concerns:**

["NO or VERY MINOR ethics concerns only"]

**Final Justification:**

The paper proposes an interesting line of work and provides valuable insights into PPO training. My initial concerns focused on three points: clarity, the optimizer selection, and the exclusive reliance on PPO to validate IIF. The authors’ rebuttal satisfactorily clarified my questions. Their additional experiments showed robust performance gains with a different optimizer choice, so I raised my score accordingly. Although evaluating IIF only with PPO remains a concern, PPO’s widespread adoption makes this limitation by itself insufficient to warrant rejection.

**Limitations:**

Yes.

**Quality:**

3

**Strengths And Weaknesses:**

Overall, the paper provides a nice extension of a data attribution method to PPO. It is well-motivated and provides a number of insights into PPO. I liked Section 4, in particular, and believe these findings to be relevant to the general RL community. If the authors address some of the mentioned weaknesses and questions satisfactorily (especially the benchmark against PPO+Adam), I am willing to raise my score.

### Strengths

- Well motivated.
- Interesting insights. I really enjoyed reading Section 4.
- Diverse set of benchmark environments.
- The authors provide code, ensuring reproducibility.
- Good presentation.

### Weaknesses

- There seems to be some confusion between online RL and on-policy RL in the paper, and these two terms seem to be used interchangeably. In my opinion, this is not correct. For example, the description under Figure 1 states that "Online reinforcement learning operates in alternating rounds of data collection and policy updates", which is a description of on-policy RL. However, DQN is an off-policy algorithm, but it is also online RL. From my perspective, the authors should replace "online RL" with "on-policy RL" to avoid confusion.
- In a similar vein, this is an RL paper, so it could use RL terminology: A record seems to be a transition, so it would be helpful to name it as such.
- The evaluation uses only a single RL algorithm (PPO), making it difficult to judge how well the framework can be adapted to other (on-policy) algorithms.
- IIF introduces a new crucial hyperparameter $p$. New hyperparameters in themselves are not a problem as long as an algorithm is reasonably robust. However, $p$ apparently must be tuned individually for each environment, which is suboptimal in practice.

---

> ### Author Rebuttal · Authors · 2025-07-29
>
> Thanks for the detailed, thoughtful and constructive feedback! We are grateful that the reviewer appreciates the contributions of our work, and address their comments in what follows.
>
> > **PPO+Adam**
>
> Thanks for raising this important point! We would like to clarify that while our IIF framework is developed for SGD, it can be used to accelerate training when the optimizer is Adam. In particular, we already used Adam in **BipedalWalker** (Fig. 6, last environment) and **RLHF** (Fig. 7) for both standard training and IIF (details in Appendices B.2 and B.3). In these cases, following [1], we train the model with Adam but compute the influence scores **as if the model is trained with SGD**, i.e., using the same formula below line 156. This approach is not perfect (we have explicitly listed it as a limitation & future work in Sec. 6), but it has already achieved substantial gain!
>
> For the first 5 environments in Fig. 6, we used SGD for both standard and IIF (details in Appendix B.2), since it already performs well in these environments. Following the reviewer’s comment, we conducted **additional experiments** on two of these environments using Adam. As the rebuttal format is text-only, we present the sample efficiency and runtime results in the table below, following Fig. 6(b).
>
> *Table*: Improvement of IIF compared to standard training in sample efficiency and runtime. Optimizer = Adam; learning rate = 3e-4, 1e-3 for MiniGrid and LunarLander, respectively.
>
> |                        | MiniGrid       | LunarLander    |
> |:-----------------------|---------------:|---------------:|
> | *SE*_ave (↑)     | 24.1 ± 1.4     | 46.7 ± 4.5     |
> | *SE*_peak (↑)    | 13.3 ± 3.1     | 62.2 ± 5.0     |
> | *RT*_peak (↑)    | 18.5 ± 1.0     | 65.9 ± 3.2     |
>
> The results show that IIF delivers a clear and substantial benefit regardless of the choice of optimizers or environments, confirming our framework’s generality. One observation is that IIF gains less with Adam compared to SGD in MiniGrid, whereas the trend is reversed for LunarLander. This is partly because Adam significantly speeds up training compared to SGD in MiniGrid (and thus reduces the room of improvement), but less so in LunarLander. We will include these new results as well as discussions on the optimizers in the revision.
>
> > **RL terminologies**
>
> 1. **Online RL vs. On-policy RL**: Thanks for highlighting the important distinction. In the revision, we will 1) add a paragraph in Sec. 3 to explicitly state that the core challenge we address—the circular dependency between data and model—is a fundamental characteristic of the entire *online RL* paradigm, while the solution we present and evaluation we perform is specific to PPO, an on-policy algorithm; 2) discuss how our framework can generalize to other on-policy and off-policy RL algorithms (see “Generalizability / extensions”)
>
> 2. **Record vs. transition**: We chose the term “record” because it more accurately reflects the unit of our data attribution method. In our framework, a record is defined as $z_i = (s_i,a_i,r_i,\log \pi_i, v_i,\hat A_i)$, which encompasses not only a raw transition but also computed values like the log probability and the advantage estimate, which are essential for influence calculation. Moreover, “record” more accurately captures the atomic unit in RLHF, which is a (prompt, response) pair. That said, we appreciate the reviewer’s suggestion and will clarify how a “record” is related to a “transition” in the revision.
>
>
> > **Generalizability / extensions**
>
> By design, our framework can generalize to other on-policy and off-policy algorithms with straightforward adaptations. The core principle of measuring influence via gradient similarity is universal. For on-policy methods, this primarily involves replacing the PPO gradient (the $\mathcal{L}^{\text{PPO}}$ term under line 156) with the loss gradient of the alternative algorithm. For value-based off-policy algorithms like DQN, we additionally need to change the target function to be the Bellman error. We will add these discussions in the revision.
>
>
> > **Clarification questions**
>
> 1. **Hyperparameter $p$**: We agree with the reviewer that an additional hyperparameter entails additional tuning; however, from our ablations in Appendix C.6, we have demonstrated that *any* level of filtering can improve performance over standard training, and that a value between 12.5% and 50% works well across settings.
>
> 2. **TracIn calculation**: We use $f^{\text{return}}$ to explain the two gradient terms in TracIn. The first is the **target gradient** which does not depend on $z_i$’s ($\nabla_\theta f^{\text{return}}(\theta_j)$), computed on a validation set and represents the desired direction for policy improvement. The second is the **per-record gradient** ($\nabla_\theta \mathcal{L}^{\text{PPO}}(\theta_j, z_i)$), computed for a single training record $z_i$ and captures its learning signal. A positive dot product between these two vectors indicates the record is helpful for policy improvement, whereas a negative score indicates it is harmful.
>
> 3. **Role of target functions**: The two target functions have different use cases. $f^{\text{action}}$ is mainly for **diagnosis**—understanding why the agent takes a specific action (e.g., abnormal / harmful) at a specific state. On the other hand, $f^{\text{return}}$ assesses contribution to **overall performance**. This makes it suitable for both analysis (as in Sec. 4.1) and algorithmic policy improvement. We believe it is not suitable to use $f^{\text{action}}$ to guide RL training because it is not tied to a *universally desirable outcome*. Using it would require an external "oracle" to determine which targeted behaviors to reinforce at each specific state, which, however, is the very problem that RL is designed to solve.
>
> > **IIF in sparse-reward settings**
>
> Our new results on MiniGrid show IIF's benefit is smaller with Adam, suggesting part of the gain may stem from compensating for SGD's inefficiencies. Nevertheless, we hypothesize that IIF could potentially be more useful in sparse-reward settings. The noisy and misleading advantage estimates common in such environments are precisely what our method identifies as "harmful records", making them ideal for filtering via IIF. Investigating the joint effects of environment and optimizer on IIF’s performance is an important direction for future work.
>
> > **Final note**
>
> If you find our response satisfactory, we kindly ask you to increase your rating to support our work. Thanks again for your time and effort in improving our work!
>
>
> **References**
>
> [1] Wang, Jiachen T., et al. "Data shapley in one training run." ICLR 2025.

---

> > ### Comment · Reviewer_1ZdL · 2025-08-04
> >
> > I thank the authors for thoroughly addressing my concerns. Their clarifications were helpful to me, particularly the additional experiments regarding the combination of PPO+Adam.
> >
> > While it would be nice to see IIF in an off-policy setting (where I agree with reviewer G83y), PPO is one of the most widely used algorithms, making the paper's insights already valuable as is.
> >
> > Therefore, I am raising my score as indicated in my review.

---

> > > ### Author Response · Authors · 2025-08-04
> > > **Thank you**
> > >
> > > Thanks for acknowledging our efforts and work! We will incorporate your feedback into the revision.

---

### Official Review · Reviewer_G83y · 2025-07-01

**Clarity:** 2
**Significance:** 4
**Originality:** 3
**Rating:** 5
**Confidence:** 4

**Summary:**

This paper operates in the on-policy reinforcement learning setting and studies local data attribution. The work proposes a gradient attribution framework that allows to identify harmful instances in a single PPO batch. It is demonstrated what structure these instances have and why they might be harmful. Then, a new algorithm that filters our harmful data points is presented and experimentally evaluated.

**Questions:**

Q1: When should I use the action or the return target function?  Why are we not using the action function for the algorithm?
Q2: Why do we need a target function that differs from the original loss. Why can’t we use the PPO objective?

**Ethical Concerns:**

["NO or VERY MINOR ethics concerns only"]

**Final Justification:**

I think this paper is an interesting attempt to solve an interesting problem and the community will benefit from knowing about it. After the discussion phase, I believe that my concerns around clarity and experimental evaluation have been addressed, and I recommend acceptance.

**Limitations:**

The work has an explicit limitations section that mentions various concerns. The text is straight-up about the limitation to gradient descent. Overall, I think this section is sufficient.

**Quality:**

3

**Strengths And Weaknesses:**

**Strengths**

**Motivation**
The problem statement is interesting and quite relevant to recent times.

**Clarity**
* The text is well structured and well written. It is easy to follow and provides sufficient detail in the method and experiment section.

**Claims and evidence**
* The claims about diagnosing learned functions and behavior formation are sufficiently well supported by sections 5.1 and 5.2.

**Method**
The method seems simple to implement and potentially quite useful.

**Novelty**
* I’m not familiar with the attribution line of work but from what I understand reading the text, the TracIn method already captures what the manuscript sets out to do. The main contribution as written in line 155 is the design of the target functions which effectively compute reference losses at the beginning of the round. This seems like a prudent choice but I believe the text here could be simplified.

**Experimental Design and Analyses**
* The structure of the experiments is well done, going from understanding the identified data points to removing the algorithmically.
* The experiments cover a sufficiently large set of domain properties.
* The qualitative experiment is appropriate for the claim of identifying and interpreting bad instances and provides nice insights into some of the learned behaviors.
* The experiments for return, sample efficiency and runtime efficiency claims are well designed, however they may lack statistical significance (see weaknesses).

__________
**Neutral Points**

**Other suggestions**
This work provides a new angle on data selection. It might make sense to reference related work in the RL area that has dealt with played with similar ideas such as experience replay techniques. These could be used as a motivation for this paper as they are generally not easily applicable to PPO.
__________

**Weaknesses**

**Clarity**
* The figures are difficult to understand. I’m not sure what direction to read them from and what they are supposed to convey. They only started making more sense once the framework was described in the text. I was confused when I first saw Figure 1 which is the opposite effect I would hope to gain from a Figure. Similarly, Figure 3 and 4 are difficult to understand, the caption is too brief and does not provide the takeaways from the plot.
* The description of online RL in section 2.1 is specific to PPO but stated as general online RL. SAC does not store log probabilities or advantage estimates. In general, the paper talks a lot about on-policy RL but calls it online RL. Some of the assumptions that lead to the algorithmic choice are specific to this wording. For instance, off-policy methods are still online but they have a data buffer on which I might be able to compute attribution scores. I believe a clearer distinction would benefit the text.
* There are a couple of statements that ought to describe the algorithm choices that are not well formulated and might need more support
  * L111: “Retraining-based methods are too costly and thus inapplicable." What makes them costly and why are they inapplicable in RL when this costlyness was not an issue in other domains?
  * L116:  “Technically ,one can compute influence scores ignoring this extra channel of influence, but these scores would deviate significantly from the ideal effect.” It is not clear to me why that is the case or how we would know this.
* It is unclear to me why the target function for agent action attribution does not have an expectation and why the validation set issue is only relevant for the return.
* It is unclear to me when I should use which target function. See Q1.
**Claims and evidence**
* The work claims to provide a principled and flexible framework but is only applied to PPO and objective seems tied to PPO.
* L137: “unlike supervised learning with a fixed validation set, the data distribution in online RL is inherently policy dependent.”  This issue is still an issue because the reference distribution is per run and differs across algorithm executions. It is not clear to me how this would be remedied by looking at a local reference policy. It is also not clear to me how an action is sampled from a policy in the equation following L142.
* The text claims that an evolving reference is crucial for meaningful evaluation. There is no evidence provided for this statement. Why can’t I use an offline dataset that is shared across runs on which I compute my estimates?

**Experimental Design and Analyses**
* 3 random seeds are generally insufficient to draw definite conclusions especially in RL [2, 3].
* The measure of variance is not reported in Figures 5 and 6.

**Relation To Scientific Literature**
* The fact that PPO learns poor advantage values has long been known, the work should probably cite [1].
* Other work on identifying samples to train on such as prioritized experience replay have long histories and many relevant works that might be useful to cite.

[1] Are Deep Policy Gradient Algorithms Truly Policy Gradient Algorithms? Ilyas et al.
[2] Deep Reinforcement Learning that Matters. Henderson et al.
[3] How Many Random Seeds? Statistical Power Analysis in Deep Reinforcement Learning Experiments. Cédric Colas et al.
__________
**Summary**

Overall, I think this paper is well written and proposes an interesting and useful idea. The method is simple and follows related work. The experiments demonstrate some benefits but reporting 3 seeds is not sufficient. There are also various points where the text could improve in clarity that I highlighted; especially the distinction between on-policy and online RL should be made clear. That all being said, I will recommend "accept with minor adjustments for now". Especially concerns about statistical validity should be remedied for me to recommend acceptance.

---

> ### Author Rebuttal · Authors · 2025-07-29
>
> Thanks for the detailed, thoughtful and constructive feedback! We are grateful that the reviewer appreciates the contributions of our work, and address their comments in what follows.
>
> > **Statistical significance of experimental results**
>
> We appreciate this critical feedback, and have performed **additional experiments** in three environments to address the concern. Specifically, we follow [1] to use **5 random seeds** and report the mean $\pm$ standard error. As the rebuttal format is text-only, we present the sample efficiency and runtime results in the table below, following Fig. 6(b).
>
> *Table*: Improvement of IIF compared to standard training in sample efficiency and runtime. Experimental configs are the same as in Fig. 6(b).
>
> |                    | FrozenLake      | MiniGrid       | LunarLander     |
> |:-------------------|----------------:|---------------:|----------------:|
> | *SE*_ave (↑)   | 34.0 ± 2.0      | 65.8 ± 3.3     | 26.0 ± 1.8      |
> | *SE*_peak (↑)  | 19.2 ± 5.9      | 61.7 ± 4.1     | 39.7 ± 3.7      |
> | *RT*_peak (↑)  | 29.5 ± 2.9      | 69.1 ± 1.7     | 44.9 ± 2.5      |
>
> The results show that IIF provides substantial improvements in sample efficiency and runtime, and these gains are **statistically robust**. We will update all figures and tables in the revision using 5 random seeds.
>
> > **Q1: Role of target functions**
>
> The two target functions have different use cases. $f^{\text{action}}$ is mainly for **diagnosis**—understanding why the agent takes a specific action (e.g., abnormal / harmful) at a specific state. On the other hand, $f^{\text{return}}$ assesses contribution to **overall performance**. This makes it suitable for both analysis (as in Sec. 4.1) and algorithmic policy improvement. We believe it is not suitable to use $f^{\text{action}}$ to guide RL training because it is not tied to a *universally desirable outcome*. Using it would require an external "oracle" to determine which targeted behaviors to reinforce at each specific state, which, however, is the very problem that RL is designed to solve.
>
> > **Q2: Target function $f^{\text{return}}$**
>
> Thanks for this excellent question! Our goal is to measure a sample's influence on the agent's **true objective** (maximizing return), so we use $f^{\text{return}}$ as a clean and direct target. The PPO loss, in contrast, is a complex surrogate designed for stable training—it combines policy, value, and entropy terms. Attributing influence to this composite loss would be difficult to interpret; it is unclear whether a sample is deemed influential because it improves the policy, refines the value estimate, or simply affects exploration via the entropy term. By decoupling the optimization algorithm from the evaluation objective, our approach allows us to ask a much more direct and meaningful question: *"Does this sample help the policy achieve higher expected returns?"*
>
> > **Clarity**
>
> 1. **Figs. 1, 3, 4**: We appreciate the comment and will revise the figures and their captions for better readability.
>
> 2. **Online RL vs. On-policy RL**: Thanks for highlighting the important distinction. In the revision, we will 1) add a paragraph in Sec. 3 to explicitly state that the core challenge we address—the circular dependency between data and model—is a fundamental characteristic of the entire *online RL* paradigm, while the solution we present and evaluation we perform is specific to PPO, an on-policy algorithm; 2) discuss how our framework can generalize to other on-policy and off-policy RL algorithms (see “Generalizability / extensions”)
>
> 3. **Statements (L111, L116)**:
>
> - L111: Retraining-based methods require training the model once for each of the $N$ records being evaluated. This is computationally expensive in any setting and particularly prohibitive in RL, where a single training run involves a long sequential process that can take hours, making $N$ separate runs infeasible.
>
> - L116: As illustrated in Figure 2, a training record has two influence pathways: direct parameter updates (green arrows) and subsequent data generation (red arrows). Any attribution method that ignores the second channel captures only a partial effect, yielding scores that likely deviate significantly from the true effects.
>
> 4. **The action target function**: This target function does not require an expectation because it is designed as a diagnostic tool for a specific behavior of interest (taking an action $a$ at a state $s$). In contrast, $f^{\text{return}}$ concerns maximizing the expected return over *a distribution of states*, thus requiring a validation dataset.
>
> 5. **Role of target functions**: See “Q1: Role of target functions”
>
> 6. **Generalizability / extensions**: By design, our framework can generalize to other on-policy and off-policy algorithms with straightforward adaptations. The core principle of measuring influence via gradient similarity is universal. For on-policy methods, this primarily involves replacing the PPO gradient (the $\mathcal{L}^{\text{PPO}}$ term under line 156) with the loss gradient of the alternative algorithm. For value-based off-policy algorithms like DQN, we additionally need to change the target function to be the Bellman error. We will add these discussions in the revision.
>
> 7. **Clarifications of L137, L142**:
>
> - L137: Using reference distributions that vary across runs is a direct consequence of our **contextual, local** framework. The goal is *not* to identify universally "good" data, but rather to answer a more immediate question: "For the agent *at its current stage of training*, which of its recent experiences were most helpful or harmful for the next training update?"
>
> - L142: We sample trajectories using the reference policy $\pi^{\text{ref}}$, i.e., $a \sim \pi^{ref}(s)$.
>
> 8. **Benefit of dynamic reference**: This is related to the previous question (L137). Using a dynamic reference is a deliberate choice to measure progress that is contextual to the agent's current state. A fixed, off-distribution reference can be misleading due to distribution mismatch; for instance, a novice agent gains little from a dataset sampled from an expert policy which covers states it never reaches.
>
>     To demonstrate this empirically, we performed **additional experiments** of single-round interventions (as in Sec 4.3, lines 221-223) using a *fixed* reference dataset from a well-trained policy. The results show that our intervention consistently improves the target function ($\Delta f^{\text{return}} > 0$) but degrades actual test performance ($\Delta \text{return} < 0$) except for the few final rounds. This validates our claim that an evolving, on-policy reference is crucial to ensure effective intervention.
>
>
> > **Experimental design and analyses**
>
> 1. **Random seeds**: See “Statistical significance of experimental results”
>
> 2. **Measure of variance**: Thanks for the thoughtful comment; we will update the figure captions to improve clarity. The boxplots in Fig. 5 show the full data distribution (median, IQR, and extent) over 3 random seeds. The shaded areas in Fig. 6 represent the standard error, used to visualize confidence and assess the **statistical significance** of the differences between the methods [1].
>
>
> > **Related work**
>
> Thanks for pointing out the relevant literature! We would like to note that we have already compared our method with an adaptation of Prioritized Experience Replay [2] in the on-policy setup (see Appendix C.5). In the revision, we will include a more thorough discussion of this line of work, and also discuss [3] in Sec. 4.1.
>
>
> > **Final note**
>
> If you find our response satisfactory, we kindly ask you to increase your rating to support our work. We are happy to engage in any follow-up questions. Thanks again for your time and effort in improving our work!
>
>
> **References**
>
> [1] Henderson, Peter, et al. "Deep reinforcement learning that matters." Proceedings of the AAAI conference on artificial intelligence. Vol. 32. No. 1. 2018.
>
> [2] Schaul, Tom, et al. "Prioritized experience replay." arXiv preprint arXiv:1511.05952 (2015).
>
> [3] Ilyas, Andrew, et al. "Are deep policy gradient algorithms truly policy gradient algorithms." arXiv preprint arXiv:1811.02553 (2018).

---

> > ### Comment · Reviewer_G83y · 2025-08-03
> >
> > Dear authors, thank you for the meticulous explanations of the role of the different target functions.
> >
> > However, I don't think all my concerns have been addressed.
> > * In general, I am not convinced that the results over 5 seeds are significantly more robust than those over 3 seeds. I maintain that the manuscript would benefit from more rigorous statistical analysis. The rebuttal refers to [1] for the choice of random seeds but [1] explicitly demonstrates the pitfalls of only using 5 seeds. It is probably also best to be consistent and use statistically robust evaluation across all environments. Including bootstrapped confidence intervals often goes a long way.
> > * I agree with Reviewer 6kkY about the complexity of the environments.
> > * Finally, both Reviewer 1ZdL and I had concerns about application to PPO only. If the claim is that a general purpose is provided, then that needs to be demonstrated and not only be argued in textual form. This is important because it is completely unclear whether this framework only works for PPO because of the subtleties underlying PPO specifically. If the paper is accepted, I encourage the authors to at least loosen the language around this.
> > * Finally, I can only evaluate what I can read unfortunately and given the abundance of clarity concerns I had, I cannot easily judge whether they will be fixed in a final manuscript.
> >
> > Overall, I would not be upset if this paper was published because I do think it addresses an interesting problem with an interesting approach. However, I am not willing to argue strong acceptance of champion the paper given its easily addressable weaknesses.

---

> ### Author Response · Authors · 2025-08-03
>
> Thank you for the response.
>
> > Random seeds
>
> We point out that our experiments follow exactly the same practice as in [1]. Specifically, in their "Experimental Analysis" section, the authors state: “To ensure fairness we run **five** experiment trials for *each* evaluation, each with a different preset random seed (all experiments use the **same set of** random seeds)...All results (including graphs) show **mean and standard error** across random seeds.” While they also discuss alternative approaches such as power analysis, their own empirical evaluation relies on five seeds.
>
> Furthermore, we wish to clarify a potential misinterpretation. Based on our reading, [1]’s primary caution is not against using five seeds *per se*, but against the lack of reproducibility when comparing results derived from **different sets of seeds**. Our work strictly adheres to their recommendation by using a **consistent set of seeds** for all comparisons.
>
> Ultimately, we believe the statistical significance of our method's improvement is clearly demonstrated by the results. **We invite all reviewers and the AC to examine Figure 6(b) as well as the new results in our response to Reviewer G83y, where the mean performance and standard error clearly demonstrate a significant gain over the baseline.**
>
>
> > Complexity of the environment
>
> Reviewer 6kkY has already acknowledged that our response and newly added experiments resolved their question.
>
> > Application to PPO
>
> As stated in the abstract and throughout the paper, our work is intentionally focused on the “widely used Proximal Policy Optimization (PPO) algorithm”. Our goal is to provide a deep and rigorous analysis for a highly relevant, state-of-the-art algorithm, rather than a shallow evaluation across multiple methods.
>
> While our framework can be easily extended to other online RL algorithms (as discussed in our previous response), we believe that demonstrating this on older methods like TRPO or DQN would add limited value. PPO’s prevalence makes it the most impactful testbed for a foundational study. We note that this focused approach is common in the literature. For example, the foundational work on influence function [2] introduced its methodology for understanding black-box predictions but focused its evaluation solely on vision tasks. This seminal work has since been successfully applied to numerous other domains. Similarly, we aim to establish a robust method on a critical algorithm, paving the way for broader application.
>
>
> > Clarity
>
> The reviewer mentioned “abundance of clarity concerns” and unsure “whether they will be fixed”. Could the reviewer provide concrete examples of what remains unclear so we can direct our efforts effectively? We respectfully note that other reviewers found the paper’s writing to be a strength (e.g., “well written and easy to follow” and “good presentation”).
>
>
> **References**
>
> [1] Henderson, Peter, et al. "Deep reinforcement learning that matters." Proceedings of the AAAI conference on artificial intelligence. Vol. 32. No. 1. 2018.
>
> [2] Koh, Pang Wei, and Percy Liang. "Understanding black-box predictions via influence functions." International conference on machine learning. PMLR, 2017.

---

> > ### Comment · Reviewer_G83y · 2025-08-03
> >
> > Dear authors,
> >
> > let me clarify a bit more.
> >
> > Application to PPO
> > It is not about whether PPO is used for this analysis or not but about the language in the manuscript. I think that I'm maybe just having an issue with the terminology of online RL rather than on-policy RL. I don't think there is any evidence provided that this approach would be as effective in off-policy learning. While I agree that technically it's probably possible to derive similar attribution terms, whether or not that is useful is an open question; especially since whether or not a datapoint is useful in off-policy learning will depend on other factors, e.g. how old it is---data might require reevaluation (thus, the framework as presented might *not* be applicable by just changing the loss to make it work). However, this point might not be as big of a concern and is also really not the key driver of my review. I'm willing to disregard it but I strongly encourage the authors to take this point as feedback and adjust the language in the manuscript to accurately reflect the contribution.
> >
> > Clarity
> > The main issue here is not my understanding of the points but rather how and whether they can make it into the manuscript. I can only evaluate what I can read and unfortunately, there is no PDF this year that I can read to assess the points that I raised in my initial review. I have decided that I am willing to give the authors the benefit of the doubt that all changes will be accurately reflected and I will rather err on the side of optimism in the face of uncertainty here.
> >
> > Which leave us with the final and most crucial point
> >
> > Statistical power
> > I don't believe it is relevant how many seeds another paper used. What is relevant is whether the presented evidence has sufficient statistical power to support the claims. One key experimental claim is that "final performance exceeds standard training in almost every environment" (L276) but if 1 standard error is being reported and we care about let's say a 95% CI then on 4/6 tasks the final performance is likely within variance in Figure 6a. For some that is because the tasks are easy and all algorithms simply converge to a good solution which could be resolved by showing things work on harder tasks. Thus, the final performance claim is not sufficiently supported.
> >
> > The results that are highlighted in the previous response are the sample efficiency results. I believe for these, the manuscript provides sufficient evidence to support the claims. Here 5 random seeds are sufficient to see that on many of the tasks the agents learn faster.
> >
> > I would be fine if more seeds had been added additional random seeds to support the claim or if the claims around final performance are adjusted. If the authors feel that it is possible too adjust the claim and can provide me with the language they intend to use in the final manuscript, I am willing to recommend acceptance.

---

> ### Author Response · Authors · 2025-08-04
> **Thank you for upholding high standards for our work**
>
> Thanks for giving us a chance to better understand your thoughts and concerns. Below, we address them and propose changes which we will make in the revision.
>
> - **Final performance CI.** We concur with the reviewer on rigorous statistical analysis. To this end, we computed the 95% confidence interval (CI) for the **performance gain** of IIF over the standard baseline across 5 random seeds (half-width = $t_{0.975,4} \times SE$ = $2.776\times SE$). The results confirm a statistically significant improvement: the 95% CI is [22.54, 130.52] for LunarLander and [24.40, 75.99] for BipedalWalker.
>
>     **Proposed change**: 1) L277, We will change “IIF’s final performance exceeds standard training in almost every environment” to “IIF’s final performance exceeds standard training in challenging tasks”. 2) We will include 95% CI for all reported results in Figs. 5, 6, 7 in the revision.
>
> - **Online RL vs on-policy RL.**   We argue that extending our framework to off-policy RL is not merely about deriving similar attribution terms; the scores are meaningful. As noted earlier, the core of our framework is to ask: *At the agent’s current stage of training, which data were most helpful or harmful for the next update?* This principle applies regardless of whether data is on-policy or off-policy, recent or old; attribution aims to determine usefulness for learning. One important caveat here is that in on-policy RL, current training data can serve for validation, while in off-policy RL, to measure progress relevant to the current agent, we must sample fresh data from the current policy to form the validation dataset, thus sacrificing the sample efficiency of our framework.
>
>     **Proposed change**: We will add a subsection (Sec. 3.2) to discuss how our framework can generalize to other on-policy and off-policy RL algorithms, with potential caveats (which could also serve as interesting future work) as described above.
>
> We also detail the changes aimed at improving clarity, thanks to the reviewer’s suggestion:
>
> - L111: We will change “Retraining-based methods are too costly and thus inapplicable” to “Retraining-based methods require training the model once for each of the records being evaluated, which is computationally expensive in any setting and particularly prohibitive in RL.”
>
> - L116: We will change “Technically, one can compute influence scores ignoring this extra channel of influence, but … errors that may compound exponentially over rounds” to “If we compute influence scores using the original formulas from standard supervised learning, they capture only the impact on parameter updates, ignoring the extra channel of influences through future data generation. As a result, the scores may deviate significantly from the true influence we seek to measure.”
>
> - L142: We will modify the equation to ​$f^{\text{return}}(\theta)=\mathbb{E}_{\tau\sim \pi^{\text{ref}},(s,a)\sim\tau}[\log\pi(a\mid s)\hat A^{\text{ref}}(s,a)]$.
>
> - L149: We will change “This provides a stable and relevant reference that evolves with training progress, which is crucial for meaningful evaluation and thus attribution as the policy improves.” to “This is a key design choice of our contextual framework, which enables us to ask: *For the agent at its current stage of training, which recent experiences were most helpful or harmful for the next update?* Unlike a fixed, off-distribution reference that may provide misleading signals due to mismatch with the agent’s current state, our dynamic reference evolves with training, providing a stable and relevant basis for meaningful evaluation and attribution.”
>
> - L154: We will add a remark below L154 to explain the use cases of these two target functions: “The two target functions have different use cases. $f^{\text{action}}$ is mainly for *diagnosis*: understanding why the agent takes a specific action at a specific state. On the other hand, $f^{\text{return}}$ (Sec. 4.2) assesses contribution to overall performance, which makes it suitable for both *analysis* (Sec. 4.1) and *algorithmic policy improvement* (Sec. 5).”
>
> - Fig. 1: We will lead the caption with “An Illustration of on-policy RL’s alternating learning cycle (Sec. 2.1) and our local data attribution framework (Sec. 3.1)”, to clarify what the figure aims to convey and where the detailed descriptions are located. We will simplify the content in the figure and highlight the most important components for ease of understanding.
>
> - Figs. 3 & 4: We understand that a lot of information required to understand these two figures are present only in the text following the plot. We will change the order of presentation, and also expand the content in the caption.
>
> We are grateful that the reviewer upholds high standards for our work and are happy to engage in further discussions if they’d like to suggest other changes.

---

> > ### Comment · Reviewer_G83y · 2025-08-04
> >
> > I appreciate the continued and fruitful discussion. After having reflected on it, I will raise my score accordingly. One final remark, I would change "in challenging tasks" to "in challenging tasks for PPO" or "in the more challenging tasks". Otherwise, there would be another discussion point about task complexity that may be out of scope here.

---

> > > ### Author Response · Authors · 2025-08-04
> > > **Thank you**
> > >
> > > Thank you for acknowledging our efforts. In the revision, we will make sure that our conclusions are drawn with appropriate caution, avoiding overextrapolation. Thanks again for upholding high standards for our work; your feedback is invaluable in helping us improve a work we take great pride in.

---

### Official Review · Reviewer_6kkY · 2025-07-03

**Clarity:** 4
**Significance:** 3
**Originality:** 3
**Rating:** 5
**Confidence:** 4

**Summary:**

This paper proposes a local attribution framework that explains how each data sample in the replay buffer affects the learning dynamics of the target functions (reward to go, policy) based on TraceIn. This local attribution method solves the loopy data dependency issue specifically pertaining to RL. The paper also proposes a simple filtering algorithm based on the attribution method that greatly enhances RL training performance.

**Questions:**

1. Can one swap TracIn with other attribution algorithms? Will the same framework still apply?
2. Is theta the whole set of network parameter? If so, could you also analyze the per layer gradient? If only one layer's gradient suffices for attribution, it can greatly reduce the computation/storage cost.

**Ethical Concerns:**

["NO or VERY MINOR ethics concerns only"]

**Final Justification:**

I recommend 5 because the proposed method is easy to understand and use. The paper itself is also joyful to read. As I requested, they also added case studies for more complex environment like Atari and BipedalWalkers. The results make sense to me. I believe the proposed method is indeed useful for debugging and accelerating real world reinforcement learning agents training.

**Limitations:**

yes

**Quality:**

4

**Strengths And Weaknesses:**

**Strength**
1. Well written and easy to follow.
2. The proposed method is well motivated and justified by both formal explanations and experiments. The TraceIn method, though not proposed by the Author, is adapted well to the RL context. Intuitively, the gradient direction of the value/policy function is the direction for improvements. If a sample loss function gradient doesn't align well with the above direction, this sample will certainly hinders the optimization of the value/policy function. Thus, it should be removed.
3. The experiment results demonstrate great practical potential of the proposed simple training algorithm. The sample efficiency gain demonstrated in those smaller enviornments are significant. Though as I request in below, more enviornments could be helpful for understanding the limit of the proposed method.

**Weakness**
1. Having more experiments in complex games (like Atari) and robotics environments would be better. For example, in the Pong game, during early stage training, agents may fall into local minima where it freezes on one side of the screen after a few moves. Could you demonstrate that your IIF method can boost sample efficiency in Atari games like this as well? And also demonstrate empirically that those transitions filtered out are reasonable. Similar case study on robotics environment could also be helpful like in hopper, will there be falling state mistakenly valued higher during training?

---

> ### Author Rebuttal · Authors · 2025-07-29
>
> Thanks for the positive review and insightful questions! We are grateful that the reviewer appreciates the contributions of our work, and address their comments in what follows.
>
> > **Results and analyses for other games**
>
> Following the reviewer’s suggestion, we conducted **new case studies** on two complex environments in the early-to-mid stage of training.
> 1. In **Pong** (Atari), we found that bottom records filtered by IIF consist of uninformative transitions (the ball being out of play or already moving away from the agent) that receive (inaccurately) high advantage estimates. By filtering out these samples, IIF achieves significant improvement in training efficiency.
> 2. In **BipedalWalker** (Robotics), our analysis revealed bottom records where the agent was incorrectly penalized with a large negative advantage for executing a successful recovery move (e.g., applying corrective torque with a deeply bent knee (~35°) during landing or push‑off).
>
> These results show that 1) bottom records feature inaccurate advantage estimates; 2) IIF is effective, holding generally across different environments. We will include these new analyses and results in the revision.
>
> > **Extension to other attribution algorithms**
>
> Yes, our framework is general and compatible with any gradient-based data attribution algorithm. The three core components of our framework (entity of attribution, target function, attribution method) are modular and can be designed independently. This allows for the straightforward integration of other attribution methods like influence functions [1] or SGD-influence [2].
>
> We selected TracIn for this study primarily for its conceptual simplicity and empirical efficiency. We believe investigating the interplay between our framework and other attribution algorithms is a promising direction for future work.
>
> > **Gradients of selected layers**
>
> Thanks for the thoughtful suggestion. In our work, influence is computed using the gradient with respect to the full set of model parameters.
>
> While using gradients from selected layers can reduce computational costs, as explored in prior work [3,4], this approximation can introduce bias to the attribution scores (e.g., see [5]). Our work instead leverages the *ghost dot product* technique from [5]. It enables highly efficient influence calculation using the full gradient (and thus avoiding the bias of layer selection), while adding negligible computational overhead to standard training. A detailed runtime analysis is in Appendix C.7 (Table 3), where we show that the overhead from our full-gradient influence calculation is minimal.
>
> > **Final note**
>
> We are happy to engage in any follow-up questions. Thanks again for your time and effort in improving our work!
>
>
> **References**
>
> [1] Koh, Pang Wei, and Percy Liang. "Understanding black-box predictions via influence functions." International conference on machine learning. PMLR, 2017.
>
> [2] Hara, Satoshi, Atsushi Nitanda, and Takanori Maehara. "Data cleansing for models trained with SGD." Advances in Neural Information Processing Systems 32 (2019).
>
> [3] Pruthi, Garima, et al. "Estimating training data influence by tracing gradient descent." Advances in Neural Information Processing Systems 33 (2020): 19920-19930.
>
> [4] Yeh, Chih-Kuan, et al. "First is better than last for language data influence." Advances in Neural Information Processing Systems 35 (2022): 32285-32298.
>
> [5] Wang, Jiachen T., et al. "Data shapley in one training run." ICLR 2025.

---

> > ### Comment · Reviewer_6kkY · 2025-08-02
> >
> > Thank you for the clarifications. All my remaining questions have been solved and I look forward to the revised version!

---

### Official Review · Reviewer_AVXm · 2025-07-04

**Clarity:** 3
**Significance:** 3
**Originality:** 3
**Rating:** 5
**Confidence:** 4

**Summary:**

This paper studies computing data attribution in online RL: essentially finding "importance/influence" of a datapoint to the overall performance in online RL. The task of data attribution has been studied in offline learning tasks while this paper extends that to online RL. The main challenge is that in online RL there is dependency between data and policy/model which breaks assumptions in previous data attribution tasks. The solution that this paper proposes is simple: it studies PPO and simply performs data attribution in each batch of learning, viewing the entire online learning as an iteration of offline learning tasks.

The paper uses the TracIn algorithm (Pruthi et al., 2020), which measures the difference in a target function using first-order Taylor approximation and using gradient updates of a loss function (Line 94). The results are quite broad: firstly, results are presented on studying which examples have a positive and negative influence on the target function. It then proposes a filtering approach where negative influence (bottom records) are removed, and this leads to better RL performance. Finally, results are presented on a LLM task of detoxifying LLMs.

Overall, I liked this paper, and while it would have been great to have more realistic LLM experiments and more ablations, I am advocating for acceptance.

**Questions:**

- Can authors compare how the proposed filtering will compare with other approaches such as filtering prompts/inputs where the pass@k performance is too high (too hard) or too low (too easy)?

- How can learning influence of different inputs be useful for a person doing RLHF for an LLM task, which is a timely application? A discussion on this can be useful. Authors have already mentioned how they can filter out bad records, but is there any other use?

**Ethical Concerns:**

["NO or VERY MINOR ethics concerns only"]

**Final Justification:**

Reviewers have provided helpful clarification. I will keep my score.

**Quality:**

3

**Strengths And Weaknesses:**

**Strengths:**

- Important problem of detecting data attribution in online RL
- Simple proposal directly extending TracIn (simplicity is better than unnecessary novelty and complexity!)
- Broad range of results from analysis, to proposing algorithms that use it to generate better results, and results on timely LLM experiments

**Weakness**

- On the improvement side, it is not fully clear whether the gains in filtering can come due to a reason other than what is proposed. E.g., some RLHF proposals filter out "easy prompts" and this leads to improvement in RL gains. The paper compares with some other filtering approaches but these are in the Appendix (C.4 and C.5) and should be discussed more in the paper.

- On the analysis side, it is not clear what the impact of finding bottom records will be besides filtering them in the algorithm.

---

> ### Author Rebuttal · Authors · 2025-07-29
>
> Thanks for the positive review and insightful questions! We are grateful that the reviewer appreciates the contributions of our work, and address their comments in what follows.
>
> > **Explanation of the performance gain**
>
> Filtering based on data attribution scores allows us to remove samples with harmful learning signals. As summarized by Reviewer 6kkY: *“The gradient direction of the value/policy function is the direction for improvements. If a sample loss function gradient doesn't align well with the above direction, this sample will hinder optimization… Thus, it should be removed.”* This principle is powerful because attribution scores are directly tied to the target function we aim to optimize (e.g., expected return), leading to a highly targeted filtering process.
>
> Beyond this general insight, our work develops a **RL-specific** interpretation of this effect. In particular, Sec 4.1 shows that the bottom records typically feature **inaccurate advantage estimates**. Removing them thus purifies the learning signal. This gives a more concrete, though partial (as noted in Appendix C.4) explanation to why filtering in RL is helpful.
>
>
> > **Difficulty-based filtering**
>
> Thanks for this insightful point. To the best of our knowledge, difficulty-based filtering (e.g., pass@k) is primarily used to improve LLM Reasoning (RLVR), with GRPO being the most commonly adopted RL algorithm. In GRPO, advantages are normalized across multiple responses for the same prompt. Consequently, for an extremely easy or hard prompt where all responses receive similar rewards, the resulting advantage estimates are near-zero, providing little learning signal.
>
> However, this logic does not apply to PPO. We demonstrate this with **a new experiment in RLHF** where we used *reward as a proxy for difficulty* and filtered records receiving top and bottom rewards. This heuristic performed worse than random because it systematically removed data with *both* highest and lowest influence scores, thereby harming the learning process. This finding aligns with our results in Appendix C.5 for traditional RL, where an analogous heuristic using *TD error as a proxy for difficulty* also proved ineffective. Therefore, our evidence shows that while valid for GRPO, difficulty-based filtering is an ineffective heuristic for PPO. We will add this discussion to the revision.
>
> Extending our framework to RLVR with GRPO is a promising direction that we are actively working on. However, due to the many technical challenges involved, we believe this is best suitable for a dedicated follow-up work.
>
> > **More applications of our framework**
>
> Thanks for this thoughtful question. In our paper, we have demonstrated that IIF achieves a significant improvement in training efficiency (4x reduction in wall-clock time) and final reward, which we believe is of sufficient interest to RLHF practitioners. Additionally, we believe our framework holds the potential in a few other applications:
>
> 1. **Debugging and improving the reward model**: A core challenge in RLHF is that the reward model is often imperfect. Our framework can identify training records that are “harmful” to the RLHF objective and trace this back to the flaws in the reward model—e.g., over-rewarding sycophantic or verbose but unhelpful responses. Practitioners can then use this set of records to debug and recalibrate the reward model, creating a cycle of iterative improvement.
>
> 2. **Active prompt curation for efficient alignment**: The efficiency of RLHF depends heavily on the quality of prompts used for generation. Our framework can be used to identify the set of prompts that provide the most value for policy improvement at any given stage of training. This would allow practitioners to strategically curate or up-sample the most informative prompts, leading to more targeted and efficient alignment in subsequent training.
>
> > **Final note**
>
> We are happy to engage in any follow-up questions. Thanks again for your time and effort in improving our work!

---

### Note · Authors · 2025-08-11

We sincerely thank all reviewers for their constructive and insightful feedback. We have addressed all comments in our revision, including 1) extending the discussion to off-policy RL; 2) adding analysis of difficulty-based heuristics; 3) examining the choice of optimizers; and 4) providing statistical significance analyses of the reported performance gains.

We are grateful for the unanimous recognition of the contribution and significance of our work. Our work establishes the **first framework** of data attribution in the context of traditional RL, demonstrates its effectiveness in RLHF, and opens a pathway for broader applications in the evolving landscape of LLMs. In particular, we are excited about its potential to substantially improve the **data efficiency** of RL-based post-training of LLMs, enhancing their *reasoning, agentic and tool use capabilities*.

Given the strong consensus from reviewers and the breadth of potential impact across both RL and LLM research, we look forward to sharing our findings with the broad community and to inspiring new, principled approaches to improving data usage in RL, LLMs, and beyond.

---

### Decision · Program_Chairs · 2025-09-17

**Decision:**

Accept (oral)

**Comment:**

This paper introduces the first data attribution framework for online reinforcement learning, adapting existing gradient-based methods to reinforcement learning, in particular the PPO algorithm. The work was unanimously praised by reviewers for its novelty, clear motivation, and significant empirical results. The paper is very well-written, and particularly in the context of the relative dearth of diagnostic tools for understanding and resolving learning difficulties in reinforcement learning, is poised to contribute a valuable tool to the RL researcher’s toolbox. The proposed filtering algorithm, IIF, demonstrates substantial improvements in sample efficiency and performance across a diverse range of task. Reviewers expressed concerns regarding statistical robustness, the choice of optimizer, and comparisons to other heuristics; however, these concerns were addressed by the authors during a highly productive rebuttal period. The authors are encouraged to include the additional evaluations run during the rebuttal period, in particular the additional random seeds and study of the adam optimizer, in their camera-ready version.